# High-fat diet in a mouse insulin-resistant model induces widespread rewiring of the phosphotyrosine signaling network

Antje Dittmann[1,2,†], Norman J Kennedy[3,†], Nina L Soltero[1], Nader Morshed[1,2], Miyeko D Mana[1,4], Ömer H Yilmaz[1,4,5], Roger J Davis[3,6,*] & Forest M White[1,2,7,**]

## Abstract

Obesity-associated type 2 diabetes and accompanying diseases have developed into a leading human health risk across industrialized and developing countries. The complex molecular underpinnings of how lipid overload and lipid metabolites lead to the deregulation of metabolic processes are incompletely understood. We assessed hepatic post-translational alterations in response to treatment of cells with saturated and unsaturated free fatty acids and the consumption of a high-fat diet by mice. These data revealed widespread tyrosine phosphorylation changes affecting a large number of enzymes involved in metabolic processes as well as canonical receptor-mediated signal transduction networks. Targeting two of the most prominently affected molecular features in our data, SRC-family kinase activity and elevated reactive oxygen species, significantly abrogated the effects of saturated fat exposure *in vitro* and high-fat diet *in vivo*. In summary, we present a comprehensive view of diet-induced alterations of tyrosine signaling networks, including proteins involved in fundamental metabolic pathways.

**Keywords** diabetes; free fatty acids; high-fat diet; obesity; phosphoproteomics
**Subject Categories** Proteomics; Signal Transduction
**Mol Syst Biol.** (2019) 15: e8849

## Introduction

Metabolic syndrome, characterized by insulin resistance, dyslipidemia, hyperglycemia, elevated blood pressure, and hepatic steatosis, predisposes individuals to the development of a range of disorders including type 2 diabetes, cardiovascular disease, nonalcoholic fatty liver disease, and cancer (Almind *et al*, 2005; Biddinger & Kahn, 2006; Font-Burgada *et al*, 2016). A combination of environmental and genetic factors such as overnutrition, increased lipolytic activity, and adipocyte hypertrophy has been found to be important for the development of the syndrome, and insulin resistance appears to be central to the establishment of disease (Dahlman *et al*, 2017; Gustafson *et al*, 2015). Insulin resistance describes a range of molecular phenotypes in which physiological regulatory functions of insulin are suppressed or altered. In peripheral tissues such as skeletal muscle and adipose tissue, insulin regulates glucose uptake, whereas in the liver, it suppresses gluconeogenesis. In addition, insulin stimulates postprandial lipogenesis and the synthesis of glycogen and protein, but inhibits lipolysis, glycogenolysis, and protein catabolism (Saltiel & Kahn, 2001). The liver plays a central role in many of these processes, and hepatic insulin resistance is thought to contribute to glycemic dysregulation whereby insulin fails to suppress hepatic gluconeogenesis but still activates lipogenesis (Brown & Goldstein, 2008).

The underlying molecular mechanisms of hepatic insulin resistance are still incompletely defined, but impaired control of gluconeogenesis is commonly linked to attenuated insulin signaling through Akt, a canonical downstream serine/threonine kinase that modulates and controls liver metabolism through several signaling cascades of kinase-mediated protein phosphorylation (Leavens & Birnbaum, 2011; Haeusler *et al*, 2014). However, studies of the paradoxical effect of insulin on concomitant lipogenesis and gluconeogenesis have challenged this view and have led to the proposal that a non-autonomous pathway of hepatic gluconeogenesis regulation is mediated by adipocyte lipolysis (Lu *et al*, 2012; Perry *et al*, 2015; Titchenell *et al*, 2015, 2016). Evidently, glucose production in the liver underlies insulin-independent regulatory mechanisms that

1   The David H. Koch Institute for Integrative Cancer Research, Massachusetts Institute of Technology, Cambridge, MA, USA
2   Center for Precision Cancer Medicine, Massachusetts Institute of Technology, Cambridge, MA, USA
3   Program in Molecular Medicine, University of Massachusetts Medical School, Worcester, MA, USA
4   Broad Institute of Harvard and MIT, Cambridge, MA, USA
5   Department of Pathology, Massachusetts General Hospital and Harvard Medical School, Boston, MA, USA
6   Howard Hughes Medical Institute, Worcester, MA, USA
7   Department of Biological Engineering, Massachusetts Institute of Technology, Cambridge, MA, USA
    *Corresponding author. Tel: +1 508 856 6054; E-mail: roger.davis@umassmed.edu
    **Corresponding author. Tel: +1 617 258 8949; E-mail: fwhite@mit.edu
    †These authors contributed equally to this work

are controlled by excess free fatty acids (FFAs) and a subsequent increase in hepatic acetyl CoA-mediated pyruvate carboxylase activity (Ashman et al, 1972; Perry et al, 2015). In addition, obesity and insulin resistance have been associated with elevated levels of reactive oxygen species (ROS) due to enhanced fatty acid utilization in mitochondria, increased ER stress, and increased expression and activation of NADPH oxidases (NOXs; Vincent & Taylor, 2005; Houstis et al, 2006; Nakamura et al, 2009; Lefort et al, 2010; Jiang et al, 2011; Rindler et al, 2013; Furukawa et al, 2017). The ensuing oxidative stress and shift in the redox environment result in macromolecule damage but also constitute an additional signaling layer capable of modifying regulatory nodes in insulin and other signaling pathways. Under normal physiological conditions, ROS, especially hydrogen peroxide ($H_2O_2$), act as second messengers that are intricately tied into the transient regulation of ligand-induced tyrosine phosphorylation signaling (Meng et al, 2004; Rhee, 2006; Tonks, 2006; Karisch & Neel, 2013). In contrast, in obesity and insulin resistance, excess ROS have been shown to modify mitogen-activated protein kinase (JNK, p38), SRC, and IKKβ signaling, either indirectly through oxidation of phosphatases and other endogenous inhibitors, or through direct oxidation of the kinase (Adler et al, 1999; Liu et al, 2000; Storz & Toker, 2003; Kamata et al, 2005; Kemble & Sun, 2009; Tiganis, 2011; Truong & Carroll, 2013). Consequently, phosphotyrosine (pTyr) phosphatases (PTPs) have been found to differentially undergo reversible oxidation of a conserved catalytic cysteine under long-term (12 and 24 weeks) high-fat diet conditions in mice, leading to their inactivation and a rewiring of the insulin-STAT5 network without showing global effects on phosphotyrosine signaling (Gurzov et al, 2014). Likewise, blocking SRC activity with short-term treatment of the dual ABL/SRC tyrosine kinase inhibitor dasatinib led to improved glucose tolerance, and hepatic JNK1/2 ablation improved insulin sensitivity in models of diet-induced obesity (DIO; Holzer et al, 2011; Vernia et al, 2014, 2016).

Molecular and cellular models of diet-induced insulin resistance are being explored to understand the complex tissue-specific events during overnutrition that drive the progression of insulin resistance and diabetes. To that end, system-level approaches have helped to elucidate the wide-ranging alterations in protein, RNA, and metabolite levels associated with obesity, and may help open up novel avenues for therapeutic intervention (Sabidó et al, 2013; Soltis et al, 2017; Krahmer et al, 2018; Li et al, 2018). However, to date, only limited data on the fatty acid-induced changes of basal hepatic phosphotyrosine networks have been collected, oftentimes relying on antibody detection of selected phosphorylation sites. Combined with the low abundance of many tyrosine phosphorylated proteins and substoichiometric phosphorylation levels, such targeted approaches leave an important and dynamic signaling layer regulating many biological processes underexplored. To address this knowledge gap in a more comprehensive and unbiased manner, we monitored diet-induced changes in hepatic phosphotyrosine signaling in vitro and in vivo using multiplexed quantitative mass spectrometry. We investigated the role of FFAs in rewiring the tyrosine network of the rat hepatoma cell line H4IIE treated with the saturated fatty acid palmitate or monounsaturated oleate. These studies revealed a significant increase in tyrosine phosphorylation when treated with palmitate but not oleate, accompanied by features of insulin resistance. Tyrosine network

analysis of livers from a DIO model displayed very similar trends with a substantial number of phosphotyrosine sites (111 of 352) being altered in high-fat diet (HFD) conditions, while diet effects on hepatic protein content were limited to approximately 14% of all quantified proteins. Interestingly, the majority of affected tyrosines reside in metabolic enzymes involved in glycolysis, the tricarboxylic acid cycle (TCA), glycogen synthesis, and amino acid metabolism, indicating a widespread dysregulation of crucial hepatic metabolic processes. Quantification of diet-induced pTyr in liver-specific insulin receptor knockout animals (LIRKO) demonstrated that only a subset of these phosphorylation changes could be attributed to hyperinsulinemia and persistent insulin receptor signaling. Moreover, PA-altered phosphorylation of a subset of tyrosines could be reversed using the antioxidants N-acetylcysteine (NAC) and butylated hydroxyanisole (BHA) in vitro, prompting us to investigate the impact of BHA on the pTyr network and monitor phenotypic effects in mice on HFD. Consistent with the presumed effect of ROS on tyrosine kinases and phosphatases, BHA treatment led to a remarkable reversal of HFD-induced changes in the tyrosine network (160 of 377 sites) and restored glucose and insulin response. Overall, the data reveal substantial and previously unknown responses to overnutrition and saturated FFA on potential regulatory sites of critical metabolic enzymes. These changes may explain the dysregulating effects of long-term caloric excess on hallmarks of diabetes such as persistently increased gluconeogenesis, and provide additional proof for insulin-independent effects of systemically elevated FFAs.

## Results

### Palmitate induces insulin resistance in vitro and increases tyrosine phosphorylation signaling

The role of FFAs in transcriptional control has been demonstrated (Papackova & Cahova, 2015), but, to date, information on the system-level effects of FFAs on signal transduction networks is lacking. Therefore, we sought to elucidate the effects of acutely elevated FFAs on hepatic signaling networks with the goal of identifying new network nodes whose activities are modified by FFAs. We used a well-established cell-based model to analyze the effects of BSA-conjugated saturated (PA—palmitic acid) and BSA-conjugated monounsaturated (OA—oleic acid) FFAs, as well as unconjugated BSA as a control, on Akt phosphorylation, ROS, glucose 6-phosphatase (G6Pase) expression, and pTyr levels. Consistent with PA promoting an insulin resistance-like phenotype (Stabile et al, 1998; Xu et al, 2007; Nakamura et al, 2009; Egnatchik et al, 2014; Gurzov et al, 2014), treating the rat hepatoma cell line H4IIE with PA for 24 h led to an attenuated pAkt (S473) response to insulin (Figs 1A and EV1A), and increased ROS (Fig 1B) and G6Pase mRNA, relative to OA, without impairing the expected insulin-induced increase of INSR/IGF1R phosphorylation and reduction of G6Pase expression (Fig 1C). To determine the effects of PA and OA on pTyr signaling in these cells, pTyr-containing peptides were labeled with isobaric mass tags (TMT; Thompson et al, 2003) for multiplexing, enriched by a two-step protocol to reduce sample complexity, and analyzed by quantitative LC-MS/MS (Fig 1D). This analysis led to the identification and quantification of hundreds of pTyr-containing peptides

(Dataset EV1). Hierarchical clustering of pTyr levels relative to the BSA control revealed that the phenotypic responses to PA were accompanied by a dose- and time-dependent marked increase in tyrosine phosphorylation (Fig 1E). Pathway analyses of significantly changed pTyr sites showed enrichment of receptor tyrosine kinase (RTK)-linked signal transduction processes coupled to MAPK cascades (Fig 1F). In particular, phosphorylation sites on the activation loops of ERK1/2 (ERK2-T183, Y185, ERK1-Y205) and JNK1/2 (JNK1/2-T183, Y185) were elevated upon PA treatment, which was verified by Western blotting (Fig EV1B). Furthermore, these sites co-clustered with similarly increased pTyr sites of the RTK MET (Y1001), and RTK-linked downstream and adaptor proteins (NCK1-Y105, PI3K P85A-Y467 and Y580, PI3K P55G-Y199, PTPN11-Y62 and Y584, CRK-Y136, CRKL-Y132, FER-Y402 and Y715, SHC1-Y349, Y423, Dataset EV1), contributing to the enrichment of MAPK- and RTK-associated pathways in the pool of signaling nodes that were upregulated by PA but not OA. In addition, we found multiple sites on SRC, and other SFK family members that were markedly increased by PA treatment at various concentrations (Dataset EV1), including an upregulated activation loop tyrosine common to SRC, LCK, FYN, HCK, and YES1. Upregulation of the phosphorylation of the activation loop tyrosine was verified manually for each experiment (Fig EV1C). This analysis also showed that protein levels of at least two of the SFK members were not affected by the duration of treatment or FFA concentration (Fig EV1D). These data suggest that key signaling network nodes, associated with each other through direct and indirect interactions, biological functions, and/or localization patterns (Sorkin & von Zastrow, 2009; Holzer *et al*, 2011), are strongly regulated by saturated FFA-induced phosphorylation changes. Many of these phosphorylation sites belong to a group of proteins linked to SFK activity, either as direct substrates or interaction partners, or through changes in their phosphorylation status as a response to SFK inhibition (Luo *et al*, 2008; Rubbi *et al*, 2011; Ferrando *et al*, 2012; Giansanti *et al*, 2014). Accordingly, ranking proteins based on their maximal normalized fold-changes revealed significant enrichment of SFK substrates after PA treatment, but not OA treatment, in a phospho-set enrichment analysis (PSEA) of known SFK substrates such as MCM7, FER, and SHC1 (Fig 1G, Dataset EV1; Subramanian *et al*, 2005; preprint: Sergushichev, 2016).

## High-fat diet induces widespread rewiring of the hepatic phosphotyrosine network

To determine whether the *in vivo* effects of high-fat diet (HFD) on mouse livers would mimic the *in vitro* effects of saturated fatty acids on hepatocytes, we used an established DIO model in C57BL/6J mice that has been shown to mirror hepatic dysfunction and signal transduction changes observed in patients (Collins *et al*, 2004; Soltis *et al*, 2017). Using a staggered HFD start to obtain an age-matched end-point, male mice were fed a normal chow diet (NC) or HFD for 6 or 16 weeks before euthanasia, liver resection, and analysis of the hepatic proteome and pTyr changes by LC-MS/MS (Fig EV2A). Mice fed a long-term HFD showed significantly increased obesity and insulin resistance-related phenotypes, including elevated body weight, fat mass, and blood glucose, and exhibited impaired tolerance to both glucose and insulin, relative to age-matched NC-fed mice (Fig EV2B and C).

We examined diet-associated differences in pTyr levels by comparing livers of HFD-fed mice (6 and 16 weeks, data set 1) to livers of age-matched NC-fed mice to capture different temporal windows of insulin resistance progression. Data from different MS runs were aligned by including the same normalization sample in each analysis to allow for inter-run comparisons through normalization relative to that sample. These analyses revealed significantly increased pTyr levels in the liver of HFD-fed mice for a considerable number of sites (111 of 352) after 16 weeks (Fig 2A, Dataset EV2), but not after 6 weeks. However, many of the significantly increased sites at week 16 displayed slightly elevated ratios already after 6 weeks, suggesting that HFD leads to a dysregulation of pTyr levels on cellular signaling nodes that accumulate over time. This effect is largely unidirectional, as the vast majority of significantly altered pTyr peptides exhibit increased levels relative to normal chow in both HFD conditions (Fig 2B).

To validate the general trends and translatability of the *in vivo* effects of HFD on liver signaling networks, we repeated the analysis in an independent cohort of female C57BL/6J mice fed a HFD for 12 and 30 weeks and age-matched NC control animals (data set 2). The mice were fed *ad libitum* prior to euthanasia and liver resection. In agreement with the first data set (Fig 2A and B), we observed a general trend toward significantly elevated pTyr levels compared with NC-fed mice on approximately 20% of all identified sites, predominantly in animals that were fed a 12-week HFD (Figs 2C and EV2D, Dataset EV3). The 266 peptides common in both analyses included 76 out of the 111 pTyr sites that were significantly changed by diet in data set 1. The majority of them correspond to metabolic enzymes and RTK-associated signaling nodes, and exhibited comparable responses to HFD in both cohorts (Fig 2D). Taken together, we found that, within the canonical insulin signaling network, HFD resulted in increased pTyr relative to NC on the Insulin receptor (INSR) and associated adaptor proteins insulin receptor substrate 2 (IRS2), GAB1 and SHC1, as well as the PI3K p85 regulatory subunit and PLCG1. Downstream of these nodes, feeding a HFD resulted in increased p38α phosphorylation on the activation loop, whereas the other MAPKs were not reproducibly detected or not significantly affected by diet. Although phosphorylation of the canonical INSR pathway was increased, the majority of upregulated pTyr sites were found in metabolic enzymes that are involved in glycolysis/gluconeogenesis/glycogenesis (F16P1, PYC, ALDOB, PGK1, Enolase, LDHA, PGM1, GSK3α/ß), urea cycle (ASSY, ARGI1), TCA cycle (FUMH, IDHC, DHE3), amino acid catabolism (FAAA, PH4H, HPPD, MAA), the fatty acid cycle (ECHA, THIM), and the retinol (AL1A1, AK1A1) and sarcosine (M2GD, BHMT1, SAHH) pathways, suggesting an HFD-induced rewiring of enzyme activities associated with important metabolic functions, or effects on protein stability or localization of these proteins. Interestingly, the response to diet for tyrosine phosphorylation on EGFR and the non-receptor tyrosine kinases JAK2, FAK1, and LYN as well as the common SFK activation loop site showed opposite patterns between the two data sets (down in data set 1, up in data set 2). While EGFR pTyr levels in 12- and 30-week HFD (data set 2) track with protein levels, which were significantly increased (Fig 3A), FAK1, JAK2, and SFK expression did not change under HFD12 or HFD30 in these mice.

In order to assess whether pTyr level changes in data set 1 were due to altered signaling activity or protein expression, we quantified

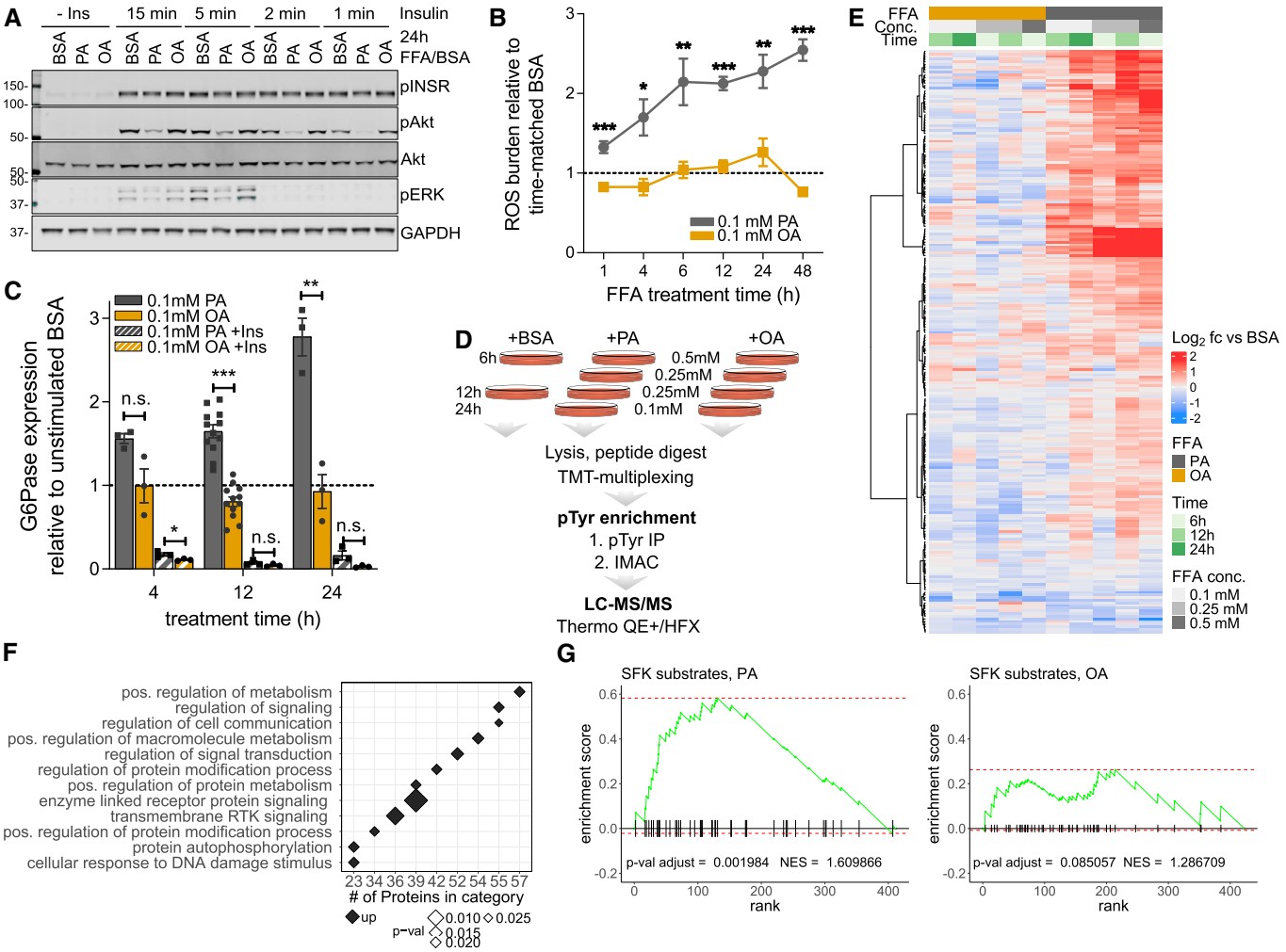

**Figure 1. Palmitate induces insulin resistance phenotype *in vitro* and profoundly perturbs signal transduction pathways.**

A   Akt phosphorylation (S473) after insulin stimulation in the presence of FFAs. Cells were treated with BSA-conjugated palmitic acid (PA), oleic acid (OA), or BSA alone (0.1 mmol/l final FFA concentration) for 24 h before being stimulated with 100 nmol/l insulin for 1–15 min. Proteins were extracted and immunoblotted with anti-pAkt-S473, anti-pERK-T202/Y204, and anti-pINSR-Y1189/90.

B   Reactive oxygen (ROS) levels after treatment with 0.1 mmol/l FFA. Cells were grown in the presence of FFA or BSA alone for indicated times, and ROS levels were measured and expressed as ratios relative to time-matched BSA (mean ± SEM; $n \geq 3$; *$P$-value < 0.05; **$P$-value < 0.01; ***$P$-value < 0.001, n.s., not significant using unpaired Student's $t$-test, two-sided).

C   Expression levels of glucose-6-phosphatase (G6Pase) after incubating cells with 0.1 mmol/l FFA or BSA alone. Three hours before the end of each time point, cells were stimulated with 100 nmol/l insulin or left unstimulated before RNA extraction. RNA levels are expressed as ratios of treatment and time-matched unstimulated BSA control (mean ± SEM; $n \geq 3$; *$P$-value < 0.05; **$P$-value < 0.01; ***$P$-value < 0.001; n.s., not significant using unpaired Student's $t$-test, two-sided).

D   Schematic workflow to study the effects of acute levels of FFAs on signal transduction pathways in hepatic cells. Rat H4IIE cells were incubated with increasing concentrations of PA, OA, or BSA alone for different time points. Phosphotyrosine peptides were prepared as described and analyzed by LC-MS/MS.

E   Unsupervised hierarchical clustering of pTyr peptide levels from cells treated with OA or PA ($n \geq 2$). Abundances are expressed as ratios ($\log_2$) relative to time-matched BSA controls.

F   Enrichment analysis of biological processes (GO-BP, DAVID) using proteins with peptides that are significantly different in PA vs. OA in any condition ($P$-value ≤ 0.05). Terms were filtered using BH-adjusted $P$-values ($P$-value ≤ 0.03).

G   PSEA to test for enrichment of SFK substrates (PhosphoSite Plus) using rank-ordered maximal normalized fold-changes for each protein.

protein levels in NC-fed and HFD-fed mice (16 weeks) by LC-MS/MS. Although HFD consumption led to a significant change of many of the identified pTyr sites, only ~14% of proteins (642 of 4,459 proteins, Fig 3B, Dataset EV4) displayed divergent expression patterns under HFD conditions relative to NC, with more proteins being downregulated in HFD16 vs. NC (392 vs. 250). Analysis of enriched GO terms in the group of significantly upregulated proteins

revealed a prevalence of lipid metabolic pathways, while the group of downregulated proteins was enriched in oxidation–reduction processes (Fig 3C). Proteins associated with lipid metabolism, transport, and ß-oxidation included apolipoproteins, fatty acid binding proteins, and mitochondrial proteins [e.g., acetyl-CoA acetyltransferase ACAT1 (THIL), acyl-CoA dehydrogenase (ACADM); Fig 3D, left panel], whereas a large number (> 20) of cytochrome P450

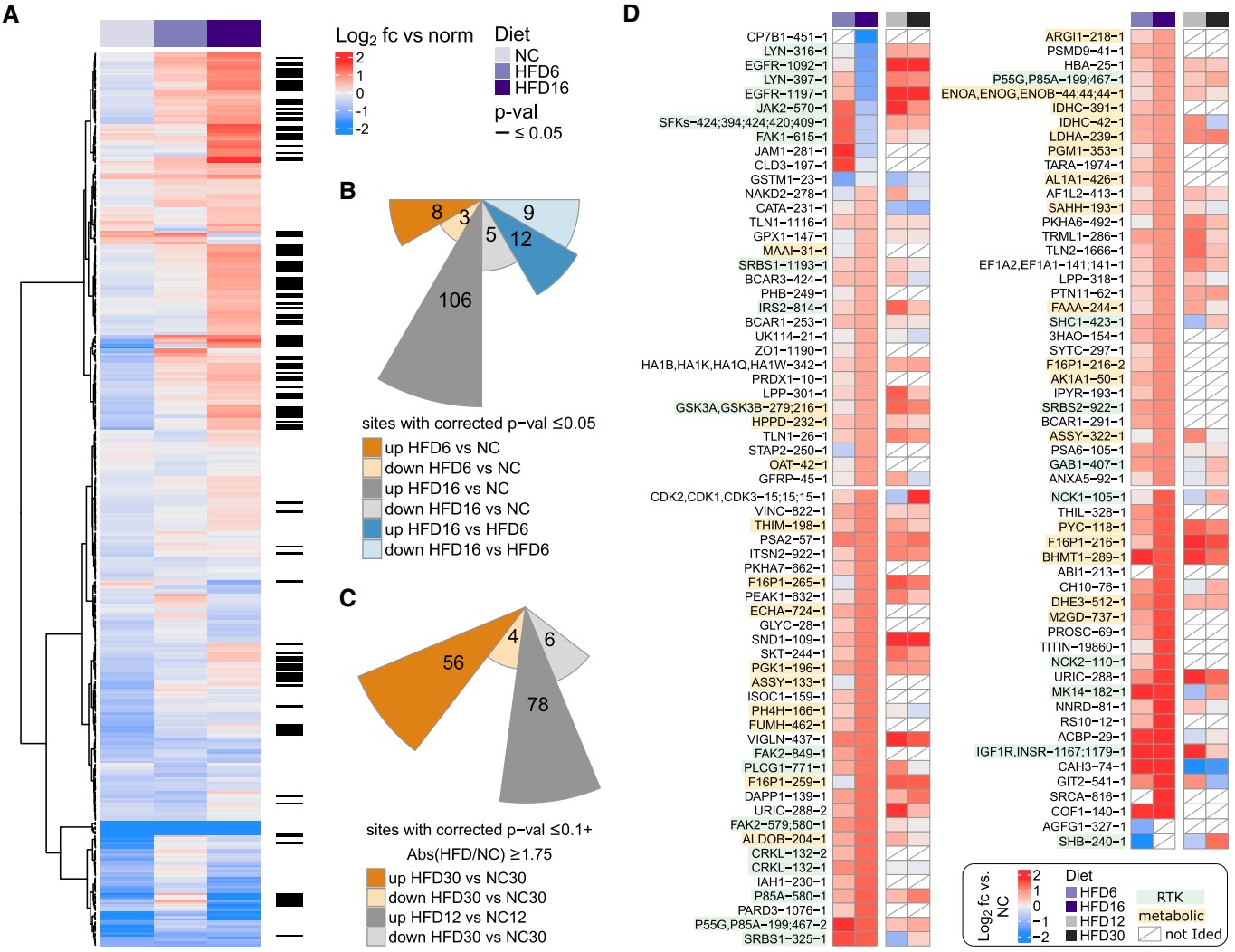

**Figure 2. HFD-induced rewiring of phosphotyrosine signaling *in vivo* affects metabolic enzymes and RTK-associated proteins.**

A   Analysis of basal phosphotyrosine levels in aged-matched NC and HFD male mice by unsupervised hierarchical clustering of peptide abundances relative to the normalization sample (log₂, data set 1). Only peptides quantified in at least two animals per condition were used. Differential regulation in any pairwise comparison was determined using a combination of moderated and standard *t*-test, and rank products. *P*-values were corrected for multiple comparisons using Benjamini-Hochberg.

B   Summary of the number of differentially regulated phosphotyrosine peptides in all pairwise comparisons plotted on a log₁₀ scale (data set 1, male mice).

C   Summary of the number of differentially regulated phosphotyrosine peptides in pairwise comparisons between NC and HFD samples from study 2 plotted on a log₁₀ scale. Abundances of peptides from female C57BL/6J mice (*n* = 2 each) on 12- or 30-week HFDs and matched NC animals were analyzed as described in data set 1 and significance was assessed as described above.

D   Abundance of peptides in HFD relative to NC (log₂) that were significantly changed in (A) (*n* = 2–4, male mice) and their HFD levels relative to age-matched NC control in data set 2 (female mice). Number after protein name indicates amino acid position of phosphorylated tyrosine in mouse protein, while number at end corresponds to peptide form (e.g., methionine oxidation, missed cleavage). White squares indicate that peptide was not identified in LC-MS/MS analysis. SFKs—SRC, LCK, YES, FYN, HCK.

family members was in the pool of proteins downregulated in HFD (Fig 3D, right panel). We assessed the general reliability of our data by benchmarking it against previously published liver proteomes (Ghose *et al*, 2011; Benard *et al*, 2016; Liu *et al*, 2017; Soltis *et al*, 2017; Krahmer *et al*, 2018). Enrichment of the top GO terms in the up- and downregulated sets are consistent with these reports, and, despite stringent protein filters which reduced the number of quantified proteins from ~5,200 to ~4,400, our protein numbers are well within the range of those studies (950–6,000).

Overall, diet-induced changes in protein abundance only poorly tracked with diet-induced changes in pTyr levels ($R^2$ = 0.25, Fig 3E)

and therefore could not fully explain the impact of diet on tyrosine signaling. Some of the most notable exceptions include fructose-1,6-bisphosphatase 1 (F16P1), FAK2, and the aforementioned acetyl-CoA acetyltransferase (THIL), which are among the 26 proteins whose expression and tyrosine phosphorylation trend concordantly in HFD16 (Fig EV3A and B), including proteins where both measures track at different magnitudes (e.g., EGFR, CYP7B, CAH3). Divergent patterns of expression and phosphorylation were observed for INSR and EF1A1, where phosphorylation was significantly up in spite of significantly decreased expression levels in HFD16 vs. NC. Taken together, these data indicate that

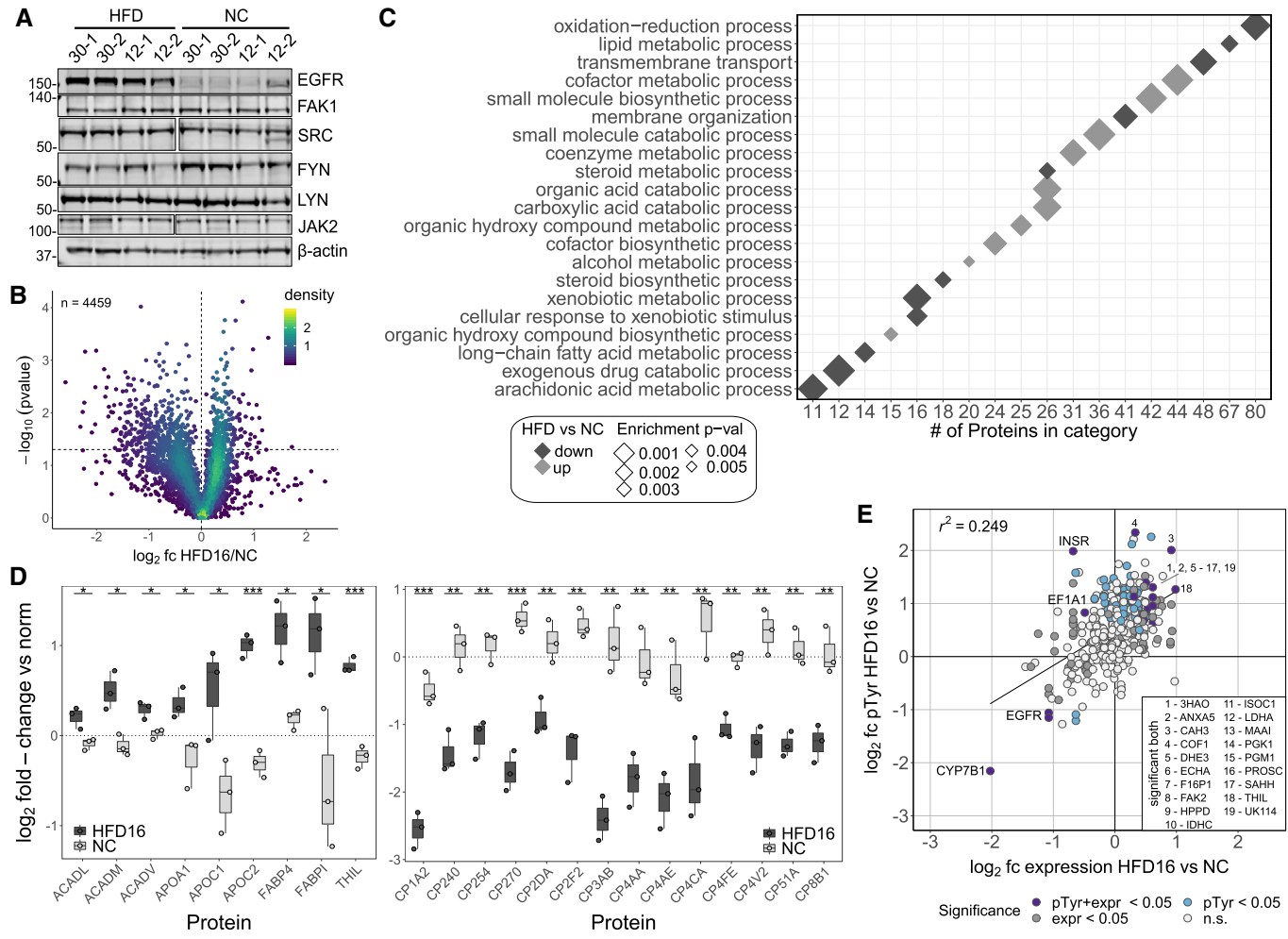

**Figure 3. HDF regulates the expression of a subset of hepatic proteins.**

A Immunoblot of total EGFR, SFKs, FAK1, and JAK2 levels in individual liver extracts in data set 2 (female mice). Lane labels correspond to duration of HFD (30 or 12 weeks) and number of replicate.

B Volcano plot of average expression ratios ($\log_2$ scale) of proteins in HDF16 relative to NC mice (male, $n = 2$–3, data set 1) and $P$-values as assed by Student's $t$-test. Two-dimensional kernel density of points was estimated per square in a grid of 100 × 100.

C Enrichment of gene ontology biological processes (DAVID GO-BP) in proteins whose expression is significantly changed by HFD16 relative to NC ($P$-value < 0.05, Fig 3B). All identified liver proteins were used as background.

D Ratios ($\log_2$) of selected proteins in livers of NC and HFD16 mice relative to normalization sample. Proteins associated with significantly enriched GO-BP terms "cofactor metabolic process" and "oxidation-reduction process" in up- and downregulated group, respectively (Fig 3B), were evaluated (median + quartile3 + 1.5*IQR/median − quartile1-1.5*IQR; $n \geq 2$; *$P$-value < 0.05; **$P$-value < 0.01; ***$P$-value < 0.001 using unpaired Student's $t$-test, two-sided).

E Scatter plot of HFD16 protein and corresponding pTyr abundances relative to NC in study 1 ($\log_2$). Data were fitted with a linear regression model. Indicated proteins display significant change in protein expression and pTyr.

overnutrition leads to a rewiring of enzymatic activity associated with metabolic signaling cascades that is only in part dependent on the impact of diet on protein content, whereas the activity of canonical RTK-signaling nodes such as EGFR, JAK2, and SFK was found to be linked to protein level changes in one or both studies.

**Hyperinsulinemia only partially explains increased tyrosine phosphorylation caused by HFD consumption**

The widespread effects of HFD consumption on basal tyrosine phosphorylation, together with reduced tolerance to glucose and insulin, prompted an investigation of the pTyr-signaling response to insulin

stimulation with the goal of defining the molecular mechanisms underlying insulin resistance in this model. Mice fed a NC diet or HFD (6 and 16 weeks) were fasted overnight and treated with insulin for 15, 30, or 60 min prior to euthanasia, liver resection, and subsequent signaling network analysis (Fig 4A). Consistent with literature reports, HFD-fed mice (6 or 16 weeks) were insulin-resistant, with significantly decreased pAkt levels following insulin stimulation relative to NC diet-fed mice (Fig 4B). At the pTyr network level, a similar response to insulin stimulation was detected across diet conditions for INSR (activation loop tyrosines), and on sites of the immediate downstream node IRS1 with respect to magnitude and time response, suggesting peak activation of these

proteins after 15 min of insulin treatment (Fig 4C). Despite this similarity in receptor activation, HFD resulted in altered insulin response signaling throughout most of the rest of the network, including on receptor proximal adaptor proteins such as IRS2, where almost all sites were dysregulated in terms of either initial response, e.g., Y734, or temporal response, e.g., Y970 (Fig 4D, Dataset EV2). Already decreased EGFR phosphorylation levels fell further after 60 min of insulin stimulation in 16 weeks of HFD-fed mice, suggesting a crosstalk between insulin signaling and the EGFR network. Further down in the network, phosphorylation of adaptor proteins including PI3K P85A (Y580, Y467), NCK1 (Y105), NCK2 (Y110), SHC1 (Y423), CRKL (Y132), GAB1 (Y407), and PLCG (Y771) was increased by 6 or 16 weeks of HFD or both, and either stayed elevated or paradoxically decreased (PLCG Y771, 6 weeks of HFD) following stimulation (Dataset EV2). Similarly, HFD-downregulated LYN sites responded positively to insulin only after 16 weeks of HFD but decreased in response to stimulation in NC-fed mice and 6 weeks of HFD-fed mice, converging to similar levels in all three diets after 60 min. The majority of metabolic network pTyr sites that were significantly increased by HFD were only responsive to insulin in the NC-fed mice; in the 6 weeks of HFD-fed mice, the response to insulin was generally higher in magnitude compared to their NC counterparts, but these sites completely failed to respond to stimulation in 16 weeks of HFD-fed mice (Fig 4E). In general, the basal 16 weeks of HFD-fed mice phosphorylation levels for those enzymes were markedly higher than their insulin-induced levels in NC, indicating not just a lack of response to insulin, but rather a diet-induced deregulation and potential activation of these pathways.

To test whether hyperinsulinemia and continuously elevated receptor signaling acted as a major driver of increased basal tyrosine phosphorylation levels, we used a liver insulin receptor knockout model (LIRKO, Fig 5A; Michael *et al*, 2000). The animals were healthy and phenocopied control mice performance on a glucose tolerance test (GTT) and weight gain (Fig EV4A and B). Male *Alb-cre Insr*$^{LoxP/LoxP}$ (LIRKO) and *Insr*$^{LoxP/LoxP}$ (Control) mice were fed a HFD (16 weeks) before analyzing hepatic pTyr levels (Fig 5B). The vast majority of sites (131 out of 157, Fig 5C, Dataset EV5) were similarly affected by HFD in the livers of LIRKO and control mice, indicating that most diet-induced alterations to pTyr signaling are independent of hepatic INSR signaling. However, liver-specific *Insr* knockout in combination with HFD did lead to decreased phosphorylation on known (IRS2-Y814, Y671, CEAM1/2-Y514/515), and potential new (AL1A1/7-Y484, ACBP-Y29, CAH3-Y74, DHE3-Y193, CLH1-Y899, Y1096) direct and indirect INSR targets and adaptor proteins, of which only a small group was increased by HFD relative to NC (Fig 5D). Loss of the insulin receptor also resulted in significant upregulation in pTyr levels for some sites after 16 weeks of HFD, including EGFR-Y1197, F16P1-Y216, PYC-Y118, and Stat3-Y705. For a subset of these proteins, we analyzed mRNA levels by qPCR and found that, with the exception of PYC (no expression change) and IRS2 (increased RNA level, decreased pTyr), significant changes in mRNA levels (Fig 5E) track with observed changes in pTyr levels. Together with decreased EGFR phosphorylation levels after 60 min of insulin treatment, these data suggest a connection between INSR signaling and the regulation of EGFR expression/activity, specifically in male mice.

## Inhibiting SFK activity and decreasing ROS burden reduces phosphotyrosine levels *in vitro*

Since treatment with saturated, but not monounsaturated FFA causes an insulin resistance phenotype with a concomitant increase in tyrosine signaling *in vitro* similar to HFD-induced phosphorylation changes detected *in vivo*, we next wanted to define and test points of intervention leading to a restoration of physiological hepatic function. Increased oxidative stress is tightly associated with obesity and insulin resistance caused by nutrient overload, inflammation, and ER stress (Houstis *et al*, 2006; Tiganis, 2011). In addition, altered redox environments have also been shown to directly affect SFK activities (Kemble & Sun, 2009; Paulsen & Carroll, 2013). To determine whether increased ROS, and consequently decreased PTP activities or SFK activation may be responsible for the altered signaling in response to FFA, we treated H4IIE hepatoma cells with OA or PA in the presence of the antioxidants N-acetyl-L-cysteine (NAC) and butylated hydroxyanisole (BHA) or the complex I inhibitor rotenone, and then subjected these samples to pTyr analysis by LC-MS/MS. Principal component analysis of the pTyr response to treatment showed that treating cells with NAC or BHA in the presence of PA moved these treatment groups closer to conditions that include OA when projected on the second coordinate axis as compared to PA alone (Fig 6A). This indicates that countering increased ROS levels aids in restoring some of the signaling networks that were altered by saturated FFA. Indeed, the thiol-containing NAC was similarly effective in reducing PA-induced tyrosine phosphorylation as the non-thiol BHA, but also showed small differences. Rotenone treatment, on the other hand, was very similar to palmitate treatment alone and did not modulate PA-induced phosphorylation at all (Fig EV5A, Dataset EV6). Focusing on the former two treatments, we were able to identify more than 30 tyrosine sites whose phosphorylation was significantly decreased by the combination of PA +BHA or PA +NAC. This set includes well-characterized SFK substrates and associated kinases (e.g., tensin, talin, calmodulin, paxillin, ABL1, FER, SHC1, PTPN11, cortactin), in addition to aspects of canonical RTK signaling (e.g., PI3K P85A, PI3K P85B, SHP-2), and MAPKs, including p38, and JNK2 (Dataset EV6). In contrast to the selective antioxidant effects, treating cells with the SRC-family/ABL kinase inhibitor dasatinib (Agostino *et al*, 2010) reversed PA-induced tyrosine phosphorylation in a concentration-dependent manner on approximately 58% of identified sites (139 out of 284, Fig EV5B, Dataset EV6). In general, the response to the inhibitor was similar in OA and PA with phosphorylation levels that were reduced to below the BSA reference in the presence of either FFA. With the exception of p38α-Y182, dasatinib failed to suppress PA-induced MAPK phosphorylation, suggesting sustained JNK and ERK activities (top cluster). Sites most strongly responding to SFK inhibition comprised several SFK sites and known substrates (e.g., paxillin, WASL, bottom clusters), whereas other known SFK substrates were aggregated into the middle cluster, which is characterized by phosphorylation levels that were reduced to levels seen in BSA in response to dasatinib. Some representatives of this cluster are enzymes involved in cell motility/cytoskeleton as well as glycolysis/gluconeogenesis, including phosphoglycerate kinase (PGK1), lactate dehydrogenase A (LDHA), and enolase (ENO), suggesting a link between changes in the SFK-related signaling machinery and metabolic reprogramming in the liver. Encouraged by the mitigating

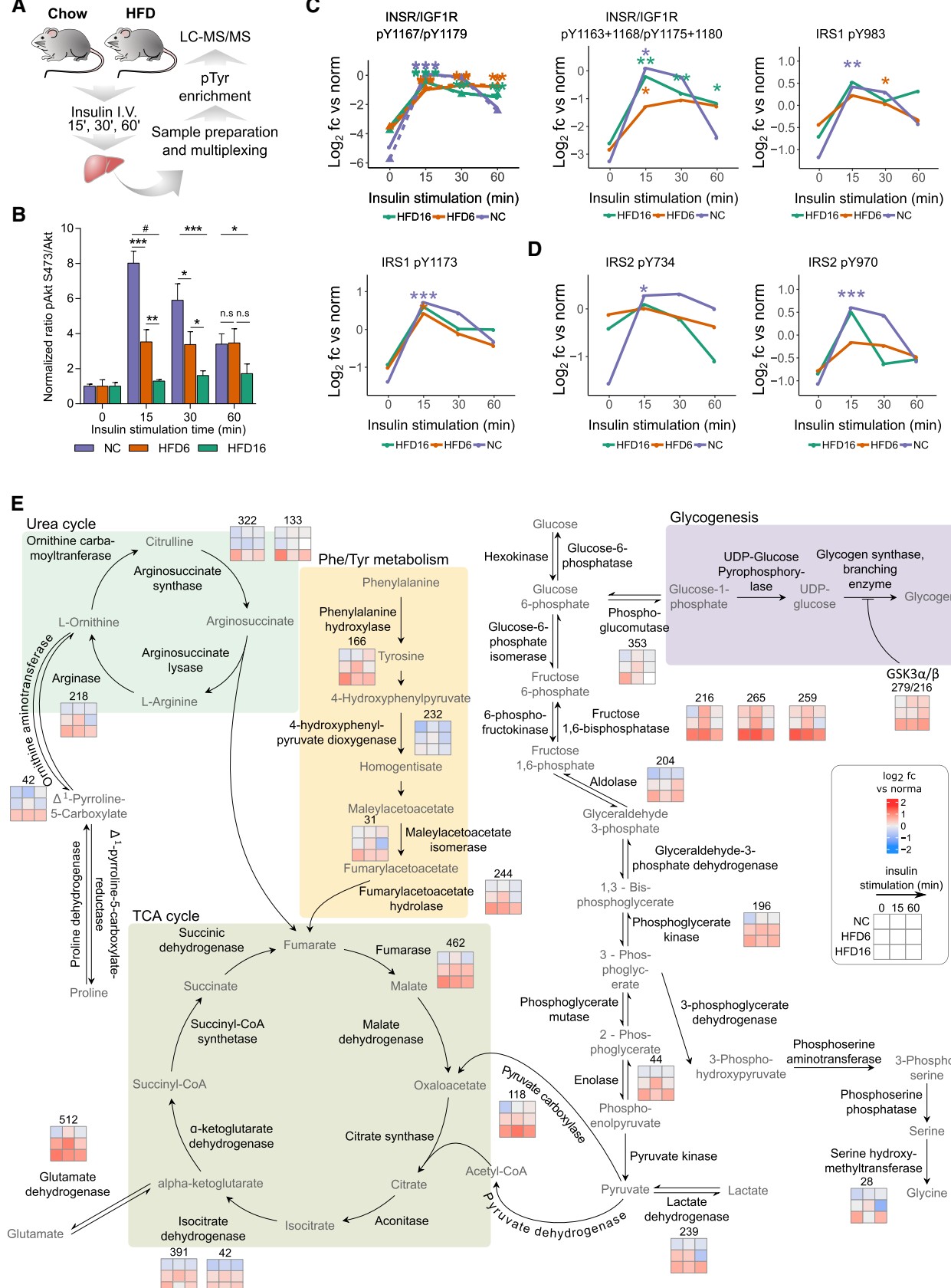

**Figure 4.**

◄

**Figure 4.  HFD alters response to insulin on proteins involved in glucose homeostasis and amino acid catabolism.**

A    Schematic workflow of effects of HFD6 and HFD16 on insulin response (data set 1). Male mice were subjected to the same diet regiment as described in Figs 2A and EV2A, and treated with insulin (10 U/kg) for 15, 30, or 60 min after overnight fasting ($n$ = 2–4). Hepatic phosphotyrosine peptide abundances were assessed relative to the same normalization sample as described in Fig 2A.

B    Immunoblot analysis of pAkt-S473 in livers from insulin-treated mice fed a NC diet or HFD for 6 or 16 weeks. pAkt levels were normalized to total Akt and expressed as normalized ratios (mean ± SEM; $n$ = 3–4; *$P$-value < 0.05; **$P$-value < 0.01; ***$P$-value < 0.005; # $P$-value < 5e-05 using unpaired $t$-test, two-sided).

C, D  Insulin response of selected pTyr peptides from INSR/IGF1R and IRS1 and 2. Average peptide abundances are reported relative to the normalization sample ($log_2$). Significance between unstimulated and stimulated peptide levels for each diet was assessed as described in Fig 2A ($n$ ≥ 2 animals; *$P$-value < 0.05; **$P$-value < 0.01; ***$P$-value < 0.001 as determined by moderated and standard $t$-test, and rank products).

E    Diagram of metabolic pathways with proteins whose tyrosine phosphorylation is changed significantly by diet, and phosphotyrosine levels relative to the normalization sample after treating animals with insulin for indicated time points. Tyrosine position in mouse protein sequence is indicated by number above quantification square.

effects on signaling, we investigated downstream responses to treating cells with NAC or dasatinib in the presence of FFA and monitored G6Pase expression and pAkt responses to insulin. Only NAC administered at high concentrations was able to reverse palmitate-induced increase in G6Pase expression (Fig 6B), but no mitigating effects of NAC, BHA, or dasatinib on the suppressed pAkt response were detected (Figs 6C and EV5C–F). Similarly, dasatinib failed to reduce ROS levels that were increased by PA (Fig EV5G). Interestingly, treating cells with high concentrations of NAC led to a decrease in insulin-induced INSR phosphorylation across all three FFA/BSA treatment conditions (Fig 6C, top panel). This appears to be in agreement with reports of the localized, transient, and reversible insulin sensitivity-promoting activities of ROS through the inactivation of negative regulators of growth factor receptor signal transduction such as PTP1B (Loh *et al*, 2009; Tiganis, 2011).

**Targeting ROS *in vivo* reverses HFD-induced increase of tyrosine phosphorylation on some but not all sites**

Due to the selective downregulating effects of BHA and NAC in the cell-based model, we were curious to see whether antioxidant properties would be effective *in vivo*, with the ultimate goal of protecting from diet-induced upregulation of tyrosine signaling. We used the same 16-week HFD model as described above, with one group of animals receiving BHA supplemented in the HFD and fed *ad libitum*, whereas the control group was fed the same HFD without BHA. LC-MS-/MS-based pTyr analysis was performed on the BHA-treated and control livers to establish the network-wide effects of BHA supplementation. Intriguingly, BHA supplementation led to a significant decrease in phosphorylation on a substantial number of pTyr sites (160 of 377, Fig 6D, Dataset EV7) relative to HFD controls, with another 36 sites that were significantly increased. In agreement with cell-based data, we saw a strong decrease of INSR phosphorylation and a concomitant decrease of CEAM1/2-Y515/514, a protein directly involved in INSR internalization (Poy *et al*, 2002) and linked to INSR activity as shown in the LIRKO mice (Fig 5C and D). The most strongly upregulated tyrosine sites belong to the super family of glutathione S-transferases (GSTPs, GSTMs), which play a major role in the detoxification of electrophilic chemicals. Comparing across treatment conditions and datasets, 46 pTyr sites were significantly altered by HFD relative to NC and were significantly affected by BHA treatment relative to the HFD control (Fig EV6A). Although almost all (39/46) of these sites were upregulated in HFD16 relative to NC and downregulated in the presence of BHA, there were seven tyrosine sites whose phosphorylation levels were

even further increased by BHA supplementation. These data suggest that BHA does not fully revert the impact of HFD and HFD-induced ROS, but instead leads to a differentially altered signaling state. To determine whether the phosphorylation changes were regulated enzymatically or by protein expression levels, we immunoblotted for a subset of proteins most strongly affected by the treatment. While changes in protein expression explain differences for CEAM1/2 and GSTM phosphorylation (McLellan *et al*, 1992), INSR and F16P1 expression levels display opposite trends as compared to their pTyr patterns (Fig 6E), indicating that these nodes are likely controlled by altered activity of kinases and/or phosphatases. Finally, to establish the phenotypic effect of BHA supplementation to HFD, we quantified starved blood glucose and tolerance to insulin and glucose. For all parameters, animals fed BHA in the diet scored significantly better than the HFD control group, suggesting that BHA supplementation offsets the negative phenotypic consequences of HFD (Figs 6F and G, and EV6B and C).

## Discussion

The development of selective insulin resistance associated with high calorie diets and obesity has been well documented (Biddinger & Kahn, 2006; Brown & Goldstein, 2008; Titchenell *et al*, 2015, 2016), but molecular features underpinning this process have been under-explored. Recently, "omics"-type data sets were generated (Sabidó *et al*, 2013; Soltis *et al*, 2017; Krahmer *et al*, 2018; Li *et al*, 2018) using *in vitro* and *in vivo* models of fatty acid- and DIO-induced insulin resistance that enable a much more comprehensive understanding of the complexity of this process by integrating genomics-, metabolomics-, and proteomics-type information. Tyrosine phosphorylation and associated signaling cascades are an important layer of regulation of cell behavior and phenotypic changes. With the exception of two recent global phosphorylation studies (Krahmer *et al*, 2018; Li *et al*, 2018), signaling modifications associated with excess lipid accumulation (e.g., triglycerides, FFAs, ceramides) have so far been focused on individual signal transduction nodes that have been found to play important roles in insulin resistance-associated inflammation, lipotoxicity, and metabolic reprogramming (Aguirre *et al*, 2000; Özcan *et al*, 2004; Fediuc *et al*, 2006; Han *et al*, 2013; Vernia *et al*, 2014, 2016). However, a systems-wide view of the impact of calorie overload on tyrosine signaling networks in insulin-response tissues has been missing.

In this study, we chose well-established *in vitro* and *in vivo* models of diet- and fatty acid-induced insulin resistance to

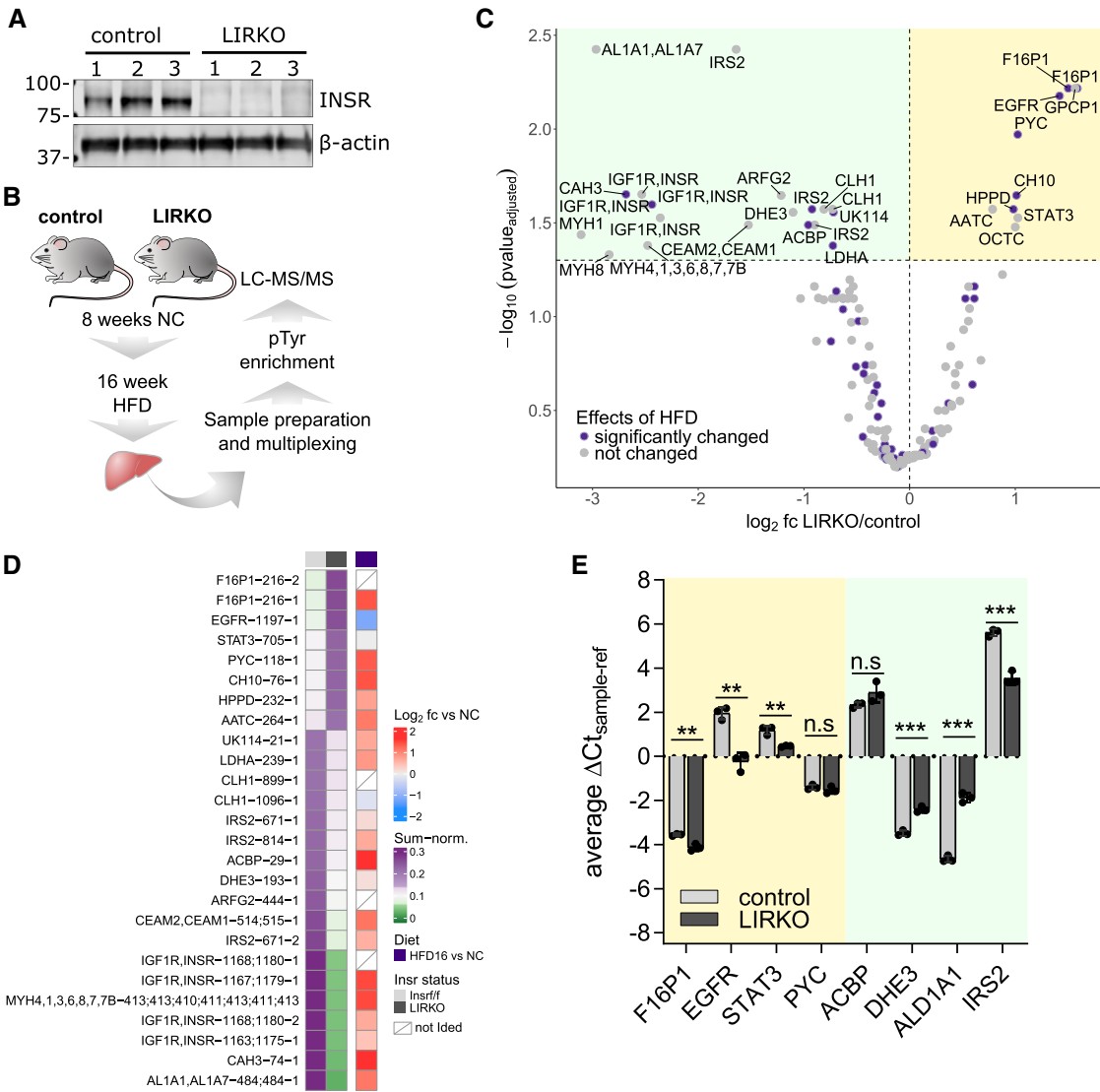

**Figure 5.  The majority of HFD-responsive pTyr sites are regulated independently of hepatic INSR expression.**

A   Immunoblot of INSR in livers of male $Insr^{LoxP/LoxP}$ mice (control) and of mice with liver-specific INSR knockout ($Alb$-$cre$ $Insr^{LoxP/LoxP}$, LIRKO).

B   Schematic representation of pTyr analysis in liver-specific INSR knockout samples (LIRKO). Male LIRKO and control mice were fed an 8-week standard chow before switching to HFD for 16 weeks as described in Fig 2A. Phosphotyrosine analysis was done as described above.

C   Volcano plot of average pTyr ratios ($\log_2$) of LIRKO ($n = 3$) and control mice ($n = 3$). Statistical significance was assessed using a combination of moderated and standard $t$-test and Benjamini–Hochberg correction. Yellow background, upregulated in LIRKO; green background, downregulated in LIRKO.

D   Heatmap of average ratios of pTyr peptides in HFD16 relative to NC (right panel) and their sum-normalized levels in LIRKO and control mice after 16 weeks of HFD (left panel). Shown are sites that were significantly (corrected $P$-value $\leq 0.05$) affected by INSR knockout in the liver. Tyrosine position in mouse sequence is indicated after the gene name followed by a number corresponding to peptide form (e.g., methionine oxidation, missed cleavage).

E   RNA levels of proteins whose pTyr levels were significantly changed in LIRKO livers relative to control. Cycle threshold (Ct) values of the gene of interest were normalized with the $C_t$ of the housekeeping gene using the delta-$C_t$ method (mean $\pm$ SEM; $n = 3$; **$P$-value $< 0.01$; ***$P$-value $< 0.001$; n.s., not significant using unpaired $t$-test). Yellow background, upregulated pTyr peptides in LIRKO; green background, downregulated pTyr peptides in LIRKO.

interrogate tyrosine phosphorylation changes in an unbiased manner using multiplexed nanoflow LC-MS/MS. We used a cell-based model of H4IIE rat hepatoma cells treated with saturated and monounsaturated fatty acids to identify signaling changes as a direct hepatic response to FFAs. This cell line is an established model that reproduces physiologically relevant insulin responses as well as lipid-associated insulin resistance phenotypes, including expression changes underlying deregulated glucose metabolism and lipotoxicity

(Yang *et al*, 2005; Wei *et al*, 2007). Consistent with features associated with insulin resistance *in vivo*, PA but not OA attenuated signaling through Akt as measured by pAkt-S473, increased mRNA levels of the gluconeogenic enzyme glucose-6-phosphatase, and led to elevated ROS levels *in vitro* (Schmoll *et al*, 2000; Puigserver *et al*, 2003). Here we examined whether saturated FFA might cause more widespread effects on cellular signaling networks that could account for transcriptional changes and the insulin-resistant phenotype. To

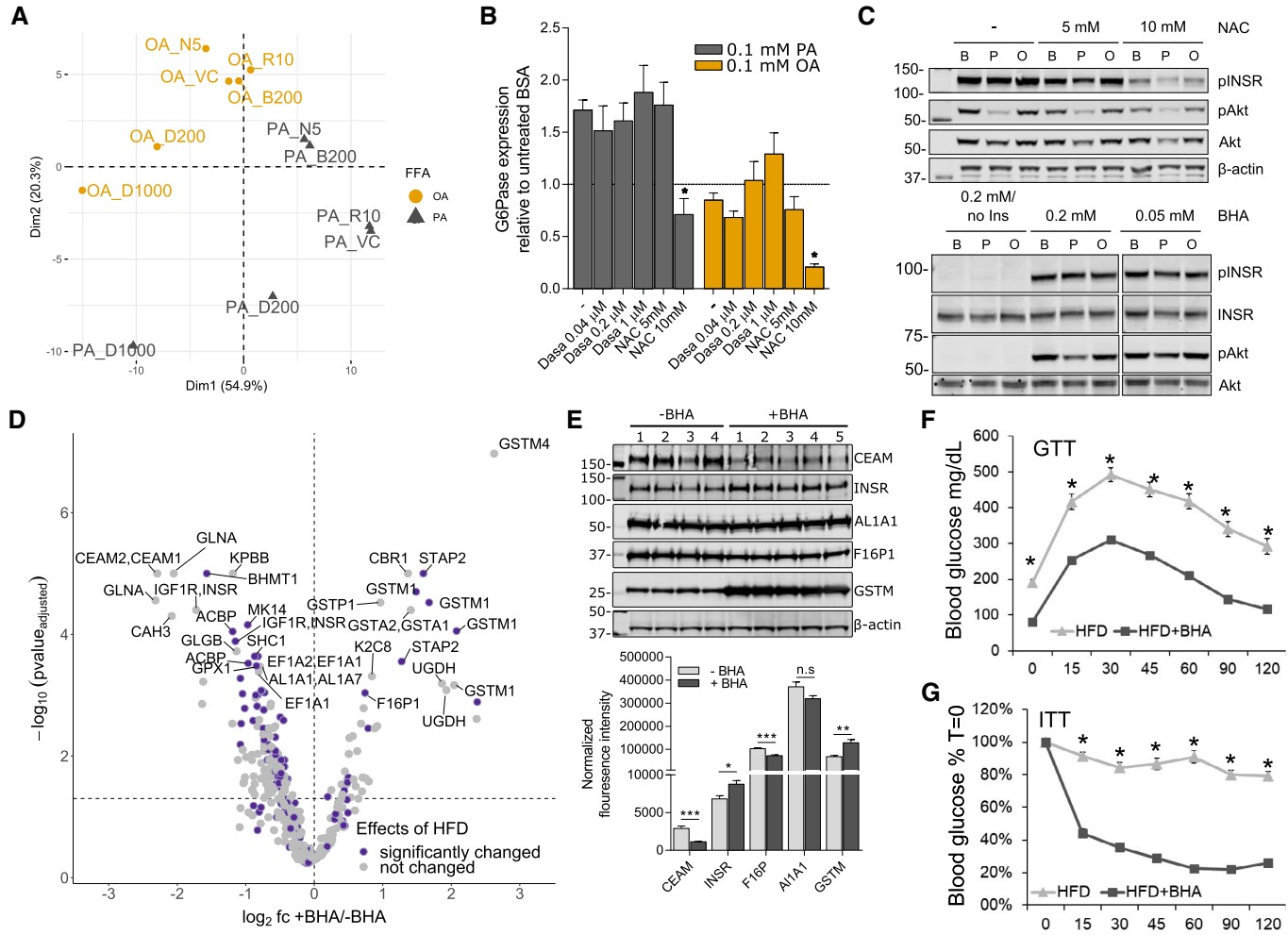

**Figure 6. Targeting ROS and SFK activity reverses FFA-induced signaling changes *in vitro* and *in vivo*.**

A   Principal component analysis of pTyr levels relative to BSA in H4IIE cells after treatment with indicated inhibitors in the presence of 0.25 mmol/l OA or PA for 12 h. OA—oleic acid, PA—palmitic acid, N5—NAC 5 mmol/l, B200—BHA 0.2 mmol/l, R10—10 nmol/l, D200—dasatinib 200 nmol/l, D1000—dasatinib 1,000 nmol/l, VC—vehicle control.

B   Expression levels of glucose-6-phosphatase (G6Pase) after incubating H4IIE cells with 0.1 mmol/l FFA or BSA for 12 h in the presence of increasing concentrations of dasatinib or NAC. RNA levels are expressed as ratios of treatment relative to untreated BSA (mean ± SEM; $n \geq 4$). Significance was assessed between FFA only and FFA + inhibitor/antioxidant (*$P$-value < 0.05 using unpaired $t$-test).

C   Representative immunoblot of Akt phosphorylation (S473) after insulin stimulation in the presence of FFAs with or without increasing concentrations of BHA or NAC. Cells were incubated with BSA or FFA media and inhibitors for 12 h and then stimulated with 100 nmol/l insulin for 5 min.

D   Volcano plot of average pTyr ratios ($\log_2$) of male HFD16 mice with or without 1.5% BHA supplementation. Mice were kept on a HFD + BHA ($n = 5$) or HFD - BHA ($n = 5$) for 16 weeks, and pTyr levels were analyzed as described above. Statistical significance was determined as described in Fig 5C. Purple indicates that these sites are differentially regulated by HFD as shown in Fig 2D.

E   Immunoblot of selected proteins with peptides that are significantly changed in HFD16 + BHA and quantification of corresponding immunoblot bands (mean ± SEM; $n = 4$–5; *$P$-value < 0.05; **$P$-value < 0.01; n.s., not significant using unpaired Student's $t$-test, two-sided).

F   Glucose tolerance test (GTT) of HFD-fed mice with or without BHA supplementation to the diet ($n = 10$ per condition; mean ± SEM; *$P$-value < 1e-03 using unpaired Student's $t$-test, two-sided).

G   Results of insulin tolerance test (ITT) of HFD-fed mice with or without BHA supplementation to the diet ($n = 10$ per condition; mean ± SEM; *$P$-value < 1e-03 using unpaired Student's $t$-test, two-sided).

this end, we employed a two-step enrichment protocol to enrich for tyrosine phosphorylated peptides, an important signal transduction mode known for its low-level abundance as compared to serine or threonine phosphorylation. This analysis revealed significant time- and concentration-dependent increases of tyrosine phosphorylation in cells treated with palmitate. Many of the proteins affected are signaling molecules that are canonical downstream effector nodes

onto which different RTK-linked signaling cascades converge, including insulin-dependent signaling. These data suggest that saturated fatty acids can induce a complex rewiring of the hepatic tyrosine network that goes beyond the JNK-mediated stress response reported so far, and seems to especially affect SFK-dependent activities. Our data show that the response to saturated fatty acids alters SFK signaling output dramatically, leading to the tyrosine

phosphorylation-mediated activation of a large number of downstream effector proteins and to changes in phenotypic outcome, in line with PA-induced changes of lipid raft clustering and SRC localization (Holzer *et al*, 2011).

We profiled protein expression and tyrosine phosphorylation in an *in vivo* model of diet-induced obesity in two independent studies that had differences in experimental design, housing, diet, sex, and genetic background. These studies were not perfectly matched in order to assess the translatability of our conclusions, and to capture the highly dynamic and adaptive nature of liver metabolism.

Expression profiling in livers of male mice (data set 1) showed that cytochrome P450 proteins were downregulated in HFD relative to NC. These data suggest that diet composition regulates the hepatic ability to metabolize specific drugs effectively, which may alter the response to therapy in affected patients. It should be noted that the expression of different members of the P450 superfamily is sex-dependent (Wiwi *et al*, 2004), which could thus have a considerable influence on diet effects, including which members are affected. Interestingly, we found that EGFR expression was differentially regulated by diet in the two studies. In data set 1 where male mice were used, HFD downregulated the expression of EGFR, whereas the opposite effect on female mice was observed in data set 2. Despite other differences in the study designs, it is tempting to speculate that the reported sex bias in the regulation of EGFR expression (Scoccia *et al*, 1991; Rando & Wahli, 2011) may translate to a differential response to caloric overload. These data warrant further investigation given EGFR's role in mediating growth-associated cues.

Our data indicate that global rewiring of the hepatic tyrosine phosphorylation network was largely reproducible despite the different experimental conditions used, and that changes in protein expression contribute to only some of the signaling alterations. The majority of reproducibly affected phosphotyrosine sites are found in proteins that are involved in important metabolic processes as well as cell motility and cytoskeletal functions. For most affected tyrosines in metabolic enzymes, functional characterization is still missing, but enzyme activity may be affected by phosphorylation either through direct activity modification or through indirect effects on protein stability or localization. Additionally, we identified sites in RTKs and associated adaptor and downstream nodes, including INSR (activation loop), IRS2, SFKs, and PI3K regulatory subunits, whose phosphorylation was upregulated by HFD, in two separate cohorts under different HFD conditions. While many of the sites were similarly regulated by the diets, some sites displayed opposite trends or were not identified across all samples, a feature common to discovery mode mass spectrometry analysis. Additional studies using targeted proteomics approaches may provide further insight into the effects of diet on those proteins.

When livers of mice treated with insulin were analyzed, we found that despite already elevated INSR phosphorylation in HFD compared to NC, the hormone was still able to elicit a strong INSR and IRS1 response in HFD mice, similar to that observed in livers of NC animals. However, phosphorylation of metabolic enzymes upregulated by diet failed to respond to insulin altogether in HFD-fed animals, suggesting a diet-induced rewiring of metabolic activities. Interestingly, in cases where these sites were affected by insulin in NC livers, the response to ligand was much more subdued compared to the effects of diet, indicative of a tightly regulated

signaling cascade that is compromised by HFD-induced signaling. Furthermore, we found that the diet-induced increase in tyrosine phosphorylation on the majority of proteins is independent of hepatic INSR expression and therefore cannot be attributed to hyperinsulinemia alone. Taken together, these data strongly suggest that the consumption of a HFD rewires and increases tyrosine-dependent signaling in a manner that is different from receptor-mediated signaling. This result supports the idea that the impact of HFD is much broader and not limited to impairing insulin's canonical regulatory functions in glucose metabolism.

Data from *in vitro* and *in vivo* models showed a remarkably widespread effect of lipid overload on the hepatic tyrosine signaling network, encompassing a variety of protein families, signaling pathways, and biological functions. We argue that these unselective effects could be due to activation of different kinase families such as SFKs or deactivation of tyrosine kinase phosphatases through increased ROS levels. Both processes have been associated with diabetes and insulin resistance *in vivo* (Houstis *et al*, 2006; Bastie *et al*, 2007; Yamada *et al*, 2010; Holzer *et al*, 2011; Gurzov *et al*, 2014). These results prompted us to explore the potential of SFK inhibitors (SFKi) and antioxidants as easily accessible strategies to mitigate the effects of insulin resistance. Inhibition of SFKs by dasatinib led to a marked reduction of tyrosine phosphorylation sites that were upregulated by palmitate in our cell-based assay. Some of the dasatinib-sensitive sites were also significantly upregulated in the *in vivo* data sets, suggesting SFK inhibition may be a valid strategy to counter diet-induced signaling changes associated with deregulated glucose homeostasis and other metabolic processes. These observations are in agreement with findings that treatment with SFKi (Holzer *et al*, 2011) or ablation of SFK gene expression causes improved glycemia in mice fed a HFD (Bastie *et al*, 2007).

The antioxidants NAC and BHA were able to reduce PA-induced phosphorylation on a subset of proteins *in vitro*, suggesting that ROS levels are directly associated with pTyr-signaling changes, most likely through inhibition of PTPs or activation of protein kinases, or potentially both. Despite overlapping sites, both antioxidants had effects distinct from each other. Previous studies have shown that NAC can exert its action by directly reacting with another molecule via a reductive thiol-disulfide exchange reaction. It thereby modulates the redox state of target proteins, potentially affecting activity, localization, or stability. The *in vitro* antioxidant properties of BHA are mostly derived from its scavenging activity of free radicals, which could explain differences in the phosphorylation response to NAC and BHA. Among the common sites, we found that phosphorylation on some SFK substrates was reduced when PA was combined with NAC or BHA. However, we failed to detect significant effects on the SFK autophosphorylation site. This could be due to the inability to associate the non-unique SFK autophosphorylation peptide with a single SFK protein by mass spectrometry. It could also indicate that SFK substrates, but not SFK activity, are regulated by ROS-linked effects, suggesting a role for ROS-mediated alterations in phosphatase activity.

Supplementing HFD with BHA *in vivo*, we found a substantial downregulation of diet-induced tyrosine phosphorylation, affecting INSR as well as known and potential novel downstream effector proteins. It is unclear whether these effects are exclusively due to antioxidant activity leading to activation of PTPs, but the data indicate that a relatively untargeted method that leads to decreased ROS

burden can globally reverse elevated tyrosine phosphorylation in a model of insulin resistance. Because BHA was added to the diet, whole-body effects that may or may not be due to alterations in hepatic signaling can be expected. Indeed, we found that BHA supplementation led to significantly improved glucose homeostasis and insulin tolerance.

Unfortunately, BHA did not simply revert the effects of HFD, but instead led to a novel signaling state, as BHA increased diet-induced tyrosine phosphorylation even further for some proteins, including FAK1 (Y614) and F16P1 (Y216, 265). While FAK1-pY614 is implicated in SRC binding (Owen *et al*, 1999), F16P1 phosphorylation sites have not been annotated functionally yet. Together with increased PTPRE-pY695 (Berman-Golan & Elson, 2007), increased FAK-pY614 can lead to increased SRC activity, yet our data indicate a decrease in SFK substrate phosphorylation on BHA supplementation. Future studies are needed to address this observation using alternative antioxidants such as Resveratrol. Taken together, BHA leads to the desired effect of mitigating diet-induced metabolic deregulation and reverses diet-rewired hepatic tyrosine network but also induces effects of unknown consequences on a post-translational and potentially transcriptional level.

In summary, we were able to show that lipid and fatty acid overload can lead to considerable rewiring of the tyrosine signal transduction networks *in vitro* and *in vivo*. These effects happen concurrently to the development of insulin resistance, and it could be speculated that changes to the signal transduction pathways in the liver are important drivers for hepatic dysfunction. In line with previous studies on tyrosine phosphatases (Gurzov *et al*, 2014) and the role of SFKs in fatty acid metabolism and insulin biology (Bastie *et al*, 2007; Holzer *et al*, 2011), we were able to show that targeting increased SFK activities or reducing ROS levels and their deactivating effect on PTPs have a significant impact on diet-rewired signaling networks and phenotypic output. Future studies should be aimed at identifying how these changes influence glucose homeostasis and the activation state of affected metabolic enzymes. Taken together, these data significantly extend the insight into molecular underpinnings of the development of diet-induced diabetes and potentially other insulin resistance-associated diseases, and provide a framework for new therapeutic hypotheses.

# Materials and Methods

## Reagents and Tools table

| Reagent/Resource | Reference or Source | Identifier or Catalog Number |
|---|---|---|
| **Experimental models** | | |
| C57BL/6J (*Mus musculus*) | Jackson Lab | Stock number 000664 |
| B6.Cg-Speer6-ps1Tg(Alb-cre)21Mgn/J (*Mus musculus*) | Jackson Lab, Postic *et al* (1999) | Stock number 003574 |
| B6.129S4(FVB)-Insrtm1Khn/J (*Mus musculus*) | Jackson Lab, Brüning *et al*, 1998 | Stock number 006955 |
| B6.129S4-Gt(ROSA)26Sortm1Sor/J (*Mus musculus*) | Jackson Lab | Stock number 003474 |
| B6.129P2-Lgr5tm1(cre/ERT2)Cle/J (*Mus musculus*) | Jackson Lab | Stock number 008875 |
| H4IIE (*Rattus norvegicus*) | Novo Nordisk | |
| **Antibodies** | | |
| pAkt-S473 | Cell Signaling Technology | 4060 |
| Total Akt | Cell Signaling Technology | 2920 |
| AL1A1 | Cell Signaling Technology | 12035 |
| CEAM1/2 | Cell Signaling Technology | 14771 |
| INSR | Cell Signaling Technology | 3020 |
| pINSR | Cell Signaling Technology | 3024 |
| EGFR | Cell Signaling Technology | 4267 |
| pERK | Cell Signaling Technology | 9101 |
| F16P1 | Cell Signaling Technology | 59172 |
| GSTM1/2/4/5 | Abcam | 178684 |
| PYC | Abcam | 128952 |
| GAPDH | Santa Cruz | 20357 |
| β-actin | Cell Signaling Technology | 4967 |
| anti-rabbit IRDye 680 | LI-COR Biosciences | 926–68023 |
| anti-rabbit IRDye 800 | LI-COR Biosciences | 926–32211 |
| anti-mouse IRDye 680 | LI-COR Biosciences | 926–68022 |

**Reagents and Tools table** (continued)

| Reagent/Resource | Reference or Source | Identifier or Catalog Number |
|---|---|---|
| anti-mouseI RDye 800 | LI-COR Biosciences | 926–32210 |
| **Oligonucleotides and sequence-based reagents** | | |
| Mouse/rat *Hprt* forward primer | This study | CTCATGGACTGATTATGGACAGGAC |
| Mouse/rat *Hprt* reverse primer | This study | GCAGGTCAGCAAAGAACTTATAGCC |
| Rat *Rplp0* forward primer | This study | GGCGACCTGGAAGTCCAACT |
| Rat *Rplp0* reverse primer | This study | GGATCTGCTGCATCTGCTTG |
| Rat *G6pc* forward primer | This study | AGCTCCGTGCCTCTGATAAA |
| Rat *G6pc* reverse primer | This study | CCCAGTATCCCAACCACAAG |
| Mouse *Egfr* forward primer | This study | GAAGCCTATGTGATGGCTAGTG |
| Mouse *Egfr* reverse primer | This study | AGGGCATGAGCTGTGTAATG |
| Mouse *Glud1* forward primer | This study | ATCGGGTGCATCTGAGAAAG |
| Mouse *Glud1* reverse primer | This study | CAGGTCCAATCCCAGGTTATAC |
| Mouse *Aldh1a1* forward primer | This study | GAGAGTGGGAAGAAAGAAGGAG |
| Mouse *Aldh1a1* reverse primer | This study | CTCATCAGTCACGTTGGAGAA |
| Mouse *Dbi* forward primer | This study | GATACATTAGGGCCAGCGTTA |
| Mouse *Dbi* reverse primer | This study | CCCAGGCACAGAGTAACAAA |
| Mouse *Pcx* forward primer | This study | CTCCGACGTGTATGAGAATGAG |
| Mouse *Pcx* reverse primer | This study | CATCTGGTTAGCCTCCACATAG |
| Mouse *Stat3* forward primer | This study | CCTTCTGCTTCGAGACAGTTAC |
| Mouse *Stat3* reverse primer | This study | AGCCTTGCCTTCCTAAATACC |
| Mouse *Fbp1* forward primer | This study | CTACGCTACCTGTGTTCTTGTG |
| Mouse *Fbp1* reverse primer | This study | AGGCAGTCAATGTTGGATGAG |
| Mouse *Irs2* forward primer | This study | CTCGGACAGCTTCTTCTTCATC |
| Mouse *Irs2* reverse primer | This study | GGATGGTCTCATGGATGTTCTG |
| **Chemicals, enzymes and other reagents** | | |
| Sequencing grade Trypsin | Promega | V5111 |
| Trypsin-EDTA (0.05%) | Thermo Fisher Scientific | 25300096 |
| MEM | Thermo Fisher Scientific | 31095029 |
| Non-essential amino acids | Thermo Fisher Scientific | 11140050 |
| Mycoplasma detection kit | Lonza | LT07-218 |
| Fetal bovine serum | Thermo Fisher Scientific | 16000044 |
| 5 μm C18, ODS-AQ, 12 nm | YMC | AQ12S05 |
| 10 μm C18, ODS-A, 12 nm | YMC | AA12S11 |
| Poros 20 MC metal chelate affinity resin | Applied Biosystems | 1542906 |
| Dasatinib | Selleckchem | S1021, lot 8 |
| Palmitic Acid | Sigma Aldrich | P5585 |
| Oleic Acid | Sigma Aldrich | O1383 |
| NaOH 1N | Sigma Aldrich | S2770 |
| Iron Chloride | Sigma Aldrich | 451649 |
| BCA assay kit | Thermo Fisher Scientific | 23225 |
| NuPAGE™ 4–12% Bis-Tris Midi Protein Gels | Thermo Fisher Scientific | WG1402BX10 |
| Nitrocellulose membrane | BioRad | 1620112 |
| ROS | Sigma Aldrich | MAK143 |
| Blotting paper | VWR | 10427805 |
| Blocking buffer | LI-COR Biosciences | 927–40000 |

**Reagents and Tools table** (continued)

| Reagent/Resource | Reference or Source | Identifier or Catalog Number |
|---|---|---|
| Prolab Isopro RMH 3000 | Purina Lab Diet | 0006973 |
| Mouse high-fat diet 1 | Bioserv | S3282 |
| Mouse high-fat diet 2 | OpenSource Diets | D12492 |
| Butylated hydroxyanisole | Sigma Aldrich | B1253 |
| Insulin Novilin R | Novo Nordisk | U-100 |
| Glucose | J.T. Baker | 1916-01 |
| TRIzol | Thermo Fisher Scientific | 15596018 |
| High Capacity cDNA Reverse Transcription kit | Thermo Fisher Scientific | 4368814 |
| RNA isolation kit | Zymo Research | R2050 |
| iQ SYBR Green Supermix | BioRad | 1708880 |
| Urea | Sigma Aldrich | U5128 |
| PhosStop | Sigma Aldrich | 4906845001 |
| Protease inh cocktail | Roche | 4693116001 |
| Dithiothreitol | Sigma Aldrich | 43819 |
| Iodoacetamide | Sigma Aldrich | I1149 |
| Triethylammonium bicarbonate buffer | Sigma Aldrich | T7408 |
| SepPak Plus | Waters | WAT020515 |
| TMT reagent | Thermo Fisher Scientific | 90406/90068 |
| PT66 | Sigma Aldrich | P3300 |
| Anti-Phosphotyrosine Antibody, clone 4G10 | Merck Millipore | 05-321 |
| Phospho-Tyrosine Mouse mAb (P-Tyr-100) | Cell Signaling Technology | 9411 |
| Protein G Plus-Agarose Suspension | Merck Millipore | IP04 |
| Sodium phosphate monobasic | Sigma Aldrich | S3139 |
| Trizma base | Sigma Aldrich | T1503 |
| Acetonitrile | Sigma Aldrich | 34998 |
| Acetic acid | Sigma Aldrich | 338826 |
| **Software** | | |
| R 3.5.1 | https://www.r-project.org/ | |
| Graph Pad Prism 6.07 | https://www.graphpad.com/ | |
| IncuCyte version 2016B software | https://www.essenbioscience.com/en/products/software/ | |
| Mascot version 2.4 | http://www.matrixscience.com/ | |
| Proteome Discoverer version 1.4.1.14 | Thermo Fisher Scientific | |
| CAMV version 1.2 | https://github.com/white-lab/CAMV | |
| **Other** | | |
| Infinite 200 Pro microplate reader | Tecan | |
| IncuCyte ZOOM live cell imager | Essen Bioscience | |
| Cell Culture microplate, 96-well, black, cearbottom | Greiner Bio-One | 655090 |
| speed vac | Thermo Fisher Scientific | |
| syringe pump | Harvard Apparatus | |
| tissue homogenizer | VWR | |
| C1000 Touch Thermo Cycler | BioRad | CFX384 |
| 100 μm capillaries | Molex, LLC - Polymicro Technologies | TSP100375 |
| 50 μm capillaries | Molex, LLC - Polymicro Technologies | TSP050375 |
| 200 μm capillaries | Molex, LLC - Polymicro Technologies | TSP200350 |

**Reagents and Tools table** (continued)

| Reagent/Resource | Reference or Source | Identifier or Catalog Number |
|---|---|---|
| FC203B fraction collector | Gilson | 171011 |
| Zorbax 300Extend-C18 5 μm 4.6 × 250 mm | Agilent | 770995-902 |
| 1200 Infinity LC systems | Agilent | |
| Orbitrap Q-Exactive Plus/Q Exactive HF-x | Thermo Fisher Scientific | |
| Easy-nLC 1000 | Thermo Fisher Scientific | |
| PicoFrit capillary column 50 μm ID × 15 cm | New Objective | PF360-50-##-N-5 |

## Methods and Protocols

### Materials

All chemicals were obtained from Sigma unless otherwise stated. 5 μm (ODS-AQ, 12 nm, AQ12S05) and 10 μm (ODS-A, 12 nm, AA12S11) reversed-phase HPLC C18 material were purchased from YMC. Poros 20 MC metal chelate affinity resin was purchased from Applied Biosystems. Dasatinib was purchased from Selleckchem.

### Cell culture

H4IIE cells were maintained in low glucose Minimum Essential Medium containing 10% FBS, 1% non-essential amino acids, and 1% L-glutamine. Cells were tested to be mycoplasma free. For short time-point treatments (< 12 h), cells were starved in medium alone for 12 h before the addition of FFA/BSA medium. Fatty acid stock solutions were prepared by coupling FFAs with bovine serum albumin (BSA). First, palmitate was dissolved in 0.01 N NaOH to final concentrations of 2–10 mmol/l while heating at 70°C for 3–5 h. Oleate was dissolved in 0.01 N NaOH and heated at 37°C. Fatty acid solution was added to equal parts of 37°C warm 10% BSA in PBS and heated at 37°C for 15 min. Inhibitor and antioxidant treatment were performed for 12 h in the presence of 0.25 mmol/l FFA before cell lysis. Cells were approximately 50% confluent before starving and treatment.

### Animals

Data set 1: We obtained C57BL/6J mice (stock number 000664), Alb-cre mice (stock number 003574; Postic et al, 1999), and InsrLoxP/LoxP mice (stock number 006955; Brüning et al, 1998) from the Jackson Laboratory. Studies were performed using male mice on the C57BL/6J background. All mice were housed in a specific pathogen-free facility accredited by the American Association for Laboratory Animal Care. We fed the mice either (i) a normal chow diet (NC, Prolab Isopro RMH 3000, Purina) for 24 weeks, or (ii) standard NC followed by 6 or 16 weeks of high-fat diet (HFD, S3282, Bioserv) until an age of 24 weeks to obtain the same age end-point. For experiments with BHA (2(3)-tert-butyl-4 hydroxyanisole), the high-fat diet contained 1.5% BHA. We measured fat and lean mass noninvasively using [1]H-MRS (Echo Medical Systems). All experiments were carried out in accordance with guidelines for the use of laboratory animals and were approved by the Institutional Animal Care and Use Committee (IACUC) of the University of Massachusetts Medical School. Mice were fasted overnight (18 h) and treated by intraperitoneal injection with insulin (Novilin R, Norvo Nordisk) for the indicated time points prior to liver resection.

Mice used in data set 2 were female C57BL/6J mice [Rosa26-LSL-LacZ (stock number: 003474) × Lgr5-eGFP-IRES-CreERT2 (stock number: 008875)]. Mice were housed in the animal facility at the Koch Institute for Integrative Cancer Research at MIT, and studies pertaining to data set 2 were approved by the MIT Institutional Animal Care and Use Committee. Mice were fed a normal chow (NC) for at least 12 weeks (Prolab RMH 3000, LabDiet) followed by 12 or 30 weeks of HFD (D12492, OpenSource Diets), with age-matched control mice staying on normal chow fed ad libitum. The mice were fed ad libitum prior to liver resection.

### GTT, ITT, and blood glucose measurements

Glucose tolerance tests (GTT) were performed by intraperitoneal injection of fasted (18 h) mice with glucose (1 g/kg). Insulin tolerance tests (ITT) were performed by intraperitoneal injection of fed mice with insulin (0.75 U/kg). Blood glucose was measured at the indicated times with an Ascensia Breeze 2 glucometer (Bayer).

### Liver resection and pulverization

Mice were euthanized at the indicated time points after an overnight fast. Livers were frozen prior to removal using clamps cooled in liquid nitrogen. The frozen livers were then pulverized into a powder using a CryoPREP impactor (Covaris) or frozen as whole organs.

### ROS measurement

Cells were cultured as described above and plated at 1,000–3,500 cells per well on black 96-well tissue plates in 4–8 replicates. They were grown overnight to a confluency of 85–95% for short time points, or to 50% confluency for time points exceeding 12 h, after which they were treated with BSA-conjugated FFA or BSA alone for 6–48 h. For ROS measurements, media was removed and replaced with fresh media without BSA and FFA, mixed with the same volume of assay buffer from a fluorometric intracellular ROS kit (MAK145, Sigma-Aldrich), and a cell-permeable ROS detection reagent, which provides readout especially for superoxide and hydroxyl radicals. Cells were imaged every 15 min using an Incu-Cyte ZOOM live cell imager (Essen Biosciences) setup with a 4× objective and dual color filter, and fluorescence was measured using IncuCyte version 2016B software over 3–4 h in culture. The measurement at which maximal fluorescence was reached, typically 1 h of incubation with ROS detection reagent, was used to compute average relative ratios of total green fluorescence intensity of FFA relative to BSA control wells.

### Isolation of RNA and protein

RNA of H4IIE cultures was isolated from an ~80% confluent well of a 6-well plate with at least three biological replicates per condition. Cells were grown for indicated times, washed twice with ice-cold PBS, and lysed in TRIzol reagent (Thermo Fisher Scientific).

Pulverized mouse livers were homogenized in TRIzol. These samples were mixed thoroughly with an equal volume of nuclease-free ethanol, and total RNA was isolated using Direct-zol RNA miniprep kit (Zymo Research) following the manufacturer's protocol, including on-column Dnase I digestion. Concentration and quality of RNA were determined using a NanoDrop One spectrophotometer (Thermo Fisher Scientific). Proteins were extracted from cells using a modified RIPA buffer (50 mmol/l HEPES, 150 mmol/l NaCl, 1% NP-40, 0.5% SDS) supplemented with protease inhibitors (HALT, Thermo Fisher Scientific). Protein concentration was measured by bicinchoninic acid (BCA) assay (Thermo Fisher Scientific).

### Quantitative real-time PCR

Total RNA (1 μg) was transcribed using the rMoMuLV-based reverse transcriptase (High Capacity cDNA Reverse Transcription kit, Applied Biosystems). SYBR-green (iQ SYBR Green Supermix, Bio-Rad) was used for the quantitative RT–PCR with the following amplification cycles: cycle of 95°C for 30 s, 40 cycles of 95°C for 15 s, and 60°C for 30 s, followed by 65°C for 5 s and a subsequent melting curve up to 95°C in increments of 0.5°C using the C1000 Touch Thermo Cycler (CFX384, Bio-Rad) operated with CFX Manager version 3.1 (Bio-Rad). Selected amplification products were further analyzed by PAGE. Up to two primers (Reagents Table) were designed for each gene of interest, and each biological replicate (three or more) was amplified in triplicate. The *Hprt* gene was used as normalization control for mouse livers and *Hprt* and *Rplp0* for H4IIE cells. Target expression in fatty acid-treated samples was calculated as ΔΔCT between internal control-normalized BSA samples and internal control-normalized FFA. Target expression of genes in LIRKO/Insr$^{f/f}$ samples was calculated as ΔCT between the gene of interest and internal loading control.

### Immunoblot analysis

Immunoblot analysis of proteins from tissue samples was done using urea homogenate as described below for MS analysis. Protein lysates from cell lines were prepared using a modified RIPA buffer as described above. Proteins were separated on a precast gradient (4–12%) Bis-Tris gel and transferred to a nitrocellulose membrane. The membrane was blocked with Odyssey blocking buffer for 1 h at room temperature and incubated with primary antibodies against pAkt-S473 (CST4060), total Akt (CST2920), AL1A1 (CST12035), CEAM1/2 (CST14771), INSR(CST3020), pINSR (CST3024), EGFR (CST4267), pERK (CST9101), F16P1 (CST59172), fibronectin (Abcam ab2413), GSTM1/2/4/5 (Abcam ab178684), and PYC (Abcam 128952) overnight at 4°C. GAPDH (sc-20357) or β-actin (CST4967) was used as loading control. The next day, the blot was incubated with secondary antibodies coupled to IRDye 680 or 800 (LI-COR) for 1 h at room temperature, and immunocomplexes were visualized using an Odyssey scanner CLx (LI-COR) and analyzed with Image Studio version 3.1 and 5.0.

### Sample preparation for MS

- Lyse-frozen mouse livers in 8 mol/l urea supplemented with 1 mmol/l sodium orthovanadate ($Na_3VO_4$), 0.1% Nonidet P-40, protease inhibitor cocktail (1 tablet/10 ml), and PhosStop using a 7 mm homogenizer unit with two short pulses at 1565–3522 × *g*.
- Wash cells with PBS and lyse in 200 μl 8 mol/l urea per 10-cm dish.

  ○ Measure protein concentration by BCA assay using the manufacturer's instruction.
  ○ Reduce disulfide bonds with 10 mmol/l dithiothreitol (DTT) in 100 mmol/l ammonium acetate for 45 min at 56°C and
  ○ Alkylate sulfhydryl groups with 55 mmol/l iodoacetamide (IAA) for 45 min at room temperature in the dark.
  ○ Dilute lysates 8-fold with 100 mmol/l ammonium acetate, pH 8.9 and digest proteins with trypsin (sequencing-grade, Promega) overnight at room temperature at a protein:trypsin ratio of 1:50 (w/w).
  ○ Stop the digestion by adding formic acid to 2% and desalt peptides using SepPak Plus cartridges and a syringe pump operated at 2 ml/min for washing and 1 ml/min for loading and eluting (solvent A: 0.1% formic acid, solvent B: 0.1% formic acid/60% ACN)
  ○ Reduce peptide volume in a speedvac concentrator, before snap freezing four hundred microgram per sample in liquid nitrogen for lyophilization. Store lyophilized peptides at −80°C.
  ○ Label peptides were labeled with TMT6/10-plex reagents in 70% ethanol/115 mmol/l triethylammonium bicarbonate (TEAB), pH 8.5 for 1 h at room temperature.
  ○ Pool samples after 1 h of labeling, bring to dryness in a speedvac, and store at −80°C until analysis.

- A normalization control corresponding to one animal on normal chow diet treated with insulin for 30 min was included in all multiplexed mass spectrometry analyses. This enabled comparison of the effects of high-fat diet on basal and insulin-responsive phosphotyrosine sites (Dataset EV2) across multiple MS runs. The effects of liver-specific insulin receptor knockout (LIRKO) were evaluated by analyzing three animals per condition in a TMT6-plex setup. Effects of butylated hydroxyanisole (BHA, Sigma-Aldrich) on phosphotyrosine levels were evaluated in a TMT10-plex setup with five animals per condition.

### Phosphopeptide enrichment

- Conjugate protein G agarose beads with 12 μg 4G10, 6 μg PY-100, and 12 μg PT-66 in 300 μl IP buffer (100 mmol/l Tris–HCl, 1% Nonidet P-40 at pH 7.4 for approximately 3 h at 4°C).
- Resuspend dried labeled peptides in 400 μl IP buffer, add to antibody beads, and incubate overnight.
- Collect beads and save supernatant (non-bound fraction) for a limited LC-MS/MS analysis of peptide levels probing peptides from the most abundant proteins. These data were used for adjustment of between-channel loading variations as described below.
- Wash beads once with IP buffer and three times with rinse buffer (100 mmol/l Tris–HCl at pH 7.4).
- Elute peptides with 100 mmol/l glycine, pH 2.5, 12.5% acetonitrile for 30 min at room temperature.
- In the meantime, load a capillary column (200 μm ID × 10 cm) in-house packed with metal chelate affinity resin with 100 mmol/l Iron(III) chloride for 30 min at a flow rate of 10 μl/min and wash with 0.1% acetic acid.
- Load eluted peptides from the phosphotyrosine IP at a flow rate of 1–2 μl/min on a pressure injection cell.

- Wash the column with 100 mmol/l NaCl, 25% acetonitrile, 1% acetic acid for 15 min, and 0.1% acetic acid for 10 min.
- Elute phosphopeptides onto an acidified trapping column (100 μm ID × 10 cm, 10 μm C18) connected in line with the $Fe^{3+}$-IMAC column using 250 mmol/l $NaH_2PO_4$, pH 8.0 at a flow rate of 1–2 μl/min.
- Rinse the column with 0.2 mol/l acetic acid (Solvent A) for 15 min before LC-MS/MS analysis as described below.

For a comprehensive quantitative global proteomics analysis, one-fourth of supernatants from two phosphotyrosine IPs containing labeled peptides from three animals on normal chow and three animals on 16-week high-fat diet (corresponding to approx. 1 mg of protein) were fractionated using an Agilent Zorbax 300Extend-C18 5 μm 4.6 × 250 mm column on an Agilent 1200 operating at 1 ml/min. Buffer A consisted of 10 mmol/l TEAB, pH 8.0 and buffer B consisted of 10 mmol/l TEAB with 99% acetonitrile, pH 8.0. Fractions were collected using a Gilson FC203B fraction collector at 1-min intervals. Samples were loaded onto the column at 1 ml/min and eluted with the following fractionation gradient: 1% B to 5% B in 10 min, 5–35% B in 60 min, ramped to 70% B in 15 min, held for 5 min before equilibrating back to 1% B. Fractions 10–90 were used for concatenation to 20 fractions. After concatenation, solvent volume was reduced in a speedvac before flash freezing and lyophilization.

### Reversed-phase nano-LC-MS/MS
LC-MS/MS of phosphotyrosine eluates was carried out as follows. Peptide separation was carried out on an Agilent 1260 (Agilent Technologies) coupled to an Orbitrap Q-Exactive Plus or Q Exactive HF-x (Thermo Scientific). The washed trapping column containing phosphotyrosine peptides from the $Fe^{3+}$-IMAC elution was connected in series with an in-house packed analytical capillary column (50 μm ID × 12 cm, 5 μm C18) with an integrated electrospray tip (1–2 μm orifice). Peptides were eluted with 70% acetonitrile in 0.2 mol/l acetic acid (Solvent B) in following gradients: 0–13% solvent B in 10 min, 13–42% in 95 min, 42–60% in 10 min, 60–100% in 5 min, and 100% for 8 min, before equilibrating back to Solvent A. The flow was split to approximately 20 nl/min. The mass spectrometer was operated in positive ion mode with a spray voltage of 2 kV and heated capillary temperature of 250°C. MS data were obtained in data-dependent acquisition mode. Full scans (MS1) were acquired in the m/z range of 350–2,000 at a resolution of 70,000 (m/z 200, QEPlus) or 60,000 (HF-X), with AGC target 3E6 and a maximum injection time of 50 ms. The top 15 most intense precursor ions were selected and isolated with an isolation width of 0.4 m/z and dynamic exclusion set to 30 s. Selected ions were HCD fragmented at normalized collision energy (NCE) 33% after accumulating to target value 1E5 with a maximum injection time of 350 ms. MS/MS acquisition was performed at a resolution of 35,000 (QEPlus) or 45,000 (HF-X). LC-MS/MS of fractionated samples was carried out on an Orbitrap Q-Exactive Plus. Peptides were separated on a PicoFrit capillary column (50 μm ID × 15 cm, Scientific Instrument Services) with an integrated electrospray tip (10 μm orifice) in-house packed with 5-μm reversed-phase HPLC C18 material (ODS-AQ, 12 nm, AQ12S05, YMC) on an Easy-nLC 1000 (Thermo). Peptides were eluted with 80% acetonitrile in 0.1% formic acid (solvent B) in following gradients: 0–10% solvent B in 5 min, 10–30% in 100 min, 30–40% in 14 min, 40–60% in 5 min, 60–100% in 2 min, and 100% for 10 min, before equilibrating back to 0.1% formic acid. All fractions were run back to

back to minimize effects from differences in instrument performance. MS data were obtained in data-dependent acquisition mode with the same parameters as described for phosphotyrosine samples.

Limited LC-MS/MS analysis of the most abundant peptides to adjust for channel-to-channel loading variation was carried out on an LTQ Orbitrap XL mass spectrometer. 100 ng of each IP supernatant was loaded onto an acidified trapping column and analyzed with gradients as follows: 0–13% solvent B in 6 min, 13–42% in 61 min, 42–60% in 7 min, 60–100% in 4 min, and 100% for 8 min, before equilibrating back to Solvent A. Full scans (MS1) were acquired in the m/z range of 350–2,000 at a resolution of 60,000 (m/z 200). The top 10 most intense precursor ions were selected and isolated with an isolation width of 3 m/z and dynamic exclusion set to 30 s. Selected ions were HCD fragmented at normalized collision energy (NCE) 75% at a resolution of 60,000. Precursors were re-isolated and CID-fragmented in the LTQ ion trap with normalized collision energy 35%.

### Peptide identification and quantification
Raw mass spectral data files were processed with Proteome Discoverer version 1.4.1.14 (DBversion: 79; Thermo Fisher Scientific) and searched against the mouse (mouse livers) or rodent (H4IIE) SwissProt database using Mascot version 2.4 (Matrix Science). TMT-reporter quantification was extracted and isotope corrected in Proteome Discoverer. MS/MS spectra were matched with an initial mass tolerance of 10 ppm on precursor masses and 20 mmu for fragment ions. Cysteine carbamidomethylation, TMT-labeled lysine, and peptide N-termini were searched as fixed modifications. Oxidized methionine and phosphorylation of serine, threonine, and tyrosine were searched as variable modifications. Phosphorylation site localization of tyrosine data was done manually using CAMV version 1.2 on all peptide spectrum matches with mascot score > 10. Peptide spectrum matches for global proteomics data were filtered by mascot score ($\geq$ 25) and precursor isolation interference (< 31%), leading to an FDR of < 0.3%. Only proteins with at least two unique peptides or three quantified peptide-to-spectrum matches (psms) were considered for further analysis.

### Data analysis
#### Data transformation
Phosphopeptide reporter ion areas were corrected for loading variations within each TMT run using the median of peptide ratios of each channel obtained from limited global proteomics analysis of IP supernatant run on the Orbitrap XL. Reporter ion areas for each phosphopeptide variant were summed and represented as ratios between each condition and the normalization sample or, in the case of LIRKO and BHA analyses, as fold-change between the averaged summed intensities of the two conditions. Global proteomics analysis was done using summed up reporter ion intensities of all unique peptides and generating the average of those intensities across the at least two animals per condition.

### Visualization and statistical analysis
Data plotting and unpaired Student's t-test followed by Holm-Sidak multiple comparisons test were performed for Western blot, qPCR, and phenotypic data using GraphPad Prism version 6.07 for Windows (GraphPad Software, www.graphpad.com). Unless otherwise stated, proteomics data were visualized and analyzed using

corresponding R (R Core Team, 2017; version 3.4.1) packages. Differential regulation for phosphoproteomics data where the number of replicates does not exceed $n = 3$ was determined using a combination of moderated $t$-test (limma, version 3.34.0; Ritchie *et al*, 2015), rank products (RankProd, version 3.4.0; Hong *et al*, 2006), and standard $t$-test (Schwämmle *et al*, 2013). Conditions with higher numbers of replicates were analyzed with a combination of standard and moderated $t$-test only. *P*-values were adjusted for multiple correction using Benjamini-Hochberg as implemented in the stats package (version 3.4.1). Significance of differences in average ($n \geq 2$ biological replicates) protein expression between HFD16 and NC was assessed by unpaired, two-tailed Student's $t$-test.

### Clustering and PCA analysis

Unsupervised hierarchical clustering using Euclidean distance and ward.D agglomeration of pTyr peptide ratios was performed using the R packages ComplexHeatmap (version 1.17.1; Gu *et al*, 2016), pvclust (2.0-0), RColorBrewer (1.1-2; Neuwirth, 2014), and ggplot2 (2.2.1; Wickham, 2009). Peptides used in the clustering were identified in at least two animals/biological replicates per condition, and phosphosite annotation for each peptide was manually validated. PCA analysis was done using the R packages pls (version 2.7-0) and factoextra (version 1.0.5; Kassambara & Mundt, 2017; Mevik *et al*, 2018).

### Protein function and pathway annotation

DAVID gene ontology analysis was performed using the R packages clusterProfiler (version 3.6.0; Yu *et al*, 2012, 2015), RDAVIDWebService (version 1.16.0, GO_BP; Fresno & Fernández, 2013), interrogating org.Mm.e.g.db (version 3.5.0), or org.Rn.e.g.db (version 3.5.0) within the Bioconductor software project (Huber *et al*, 2015). Proteins/genes from all peptides identified in the corresponding samples were used as the background list. Benjamini-Hochberg was used as the *P*-value adjustment method.

### PSEA analysis

SFK substrates were extracted from PhosphoSite Plus (Hornbeck *et al*, 2015). For each protein, maximal normalized fold-changes relative to BSA per FFA condition were used to test enrichment of these substrates in a ranked list with the R package fgsea (version 1.4.1; preprint: Sergushichev, 2016) using 10,000 permutations of the dataset and BH adjustment of *P*-values to estimate the statistical significance of the enrichment score.

## Data availability

The mass spectrometry proteomics data have been deposited to the ProteomeXchange Consortium (http://proteomecentral.proteomexchange.org) via the PRIDE partner repository (Vizcaíno *et al*, 2013) with the dataset identifiers PXD011458 (https://www.ebi.ac.uk/pride/archive/projects/PXD011458) and PXD013815 (https://www.ebi.ac.uk/pride/archive/projects/PXD013815).

Expanded View for this article is available online.

## Acknowledgements

This work was supported by a Quinquennial Koch Institute Postdoctoral fellowship (AD), NIH grants DK090963 (RJD and FMW), DK107220 (RJD), and the MIT Center for Precision Cancer Medicine (FMW). RJD is an Investigator of the Howard Hughes Medical Institute.

## Author contributions

Conceptualization, AD, NJK, RJD, FMW; Methodology, RJD, FMW; Formal Analysis, AD, NJK; Investigation, AD, NJK, NLS, NM, MDM; Writing—Original Draft, AD, RJD, FMW; Writing—Review & Editing, AD, NJK, NLS, NM, MDM, ÖHY, RJD, FMW; Visualization, AD, NJK; Funding Acquisition, RJD, FMW; Resources, ÖHY, RJD, FMW; Supervision, RJD, FMW.

## Conflict of interest

The authors declare that they have no conflict of interest.

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
