## [Review Process File · Molecular Systems Biology]

High fat diet in a mouse insulin-resistance model induces wide-spread re-wiring of the phosphotyrosine signaling network

Antje Dittmann, Norman J. Kennedy, Nina L. Soltero, Nader Morshed, Miyeko D. Mana, Ömer H. Yilmaz, Roger J. Davis, Forest M. White

Review timeline:

Submission date:	1 February 2019
Editorial Decision:	4 March 2019
Revision received:	17 May 2019
Editorial Decision:	27 June 2019
Revision received:	8 July 2019
Accepted:	9 July 2019

Editor: Maria Polychronidou

Transaction Report:

1st Editorial Decision

4 March 2019

Thank you again for submitting your work to Molecular Systems Biology. We have now heard back from the three referees who agreed to evaluate your study. Overall the reviewers appreciate that the study seems interesting and the presented datasets are likely to be a useful resource for future studies. However, as you will see below, the reviewers raise a number of concerns, which we would ask you to address in a major revision.

Without repeating all the points/comments listed below, some of the more fundamental issues that need to be convincingly addressed are the following:

- Reviewer #1 refers to the need to better account for changes at the protein level when reporting pY changes.
- Reviewer #2 raises concerns regarding the comparison of the different datasets. S/he provides constructive suggestions on how to address these issues.

All other issues raised by the reviewers need to be satisfactorily addressed. Regarding the suggestion of reviewer #3 to split the study in multiple papers, we do not think that this is necessary. As you may already know, our editorial policy allows in principle a single round of major revision so it is essential to provide responses to the reviewers' comments that are as complete as possible. Please feel free to contact me in case you would like to discuss in further detail any of the issues raised by the reviewers.

If you feel you can satisfactorily deal with these points and those listed by the referees, you may wish to submit a revised version of your manuscript. Please attach a covering letter giving details of the way in which you have handled each of the points raised by the referees. A revised manuscript will be once again subject to review and you probably understand that we can give you no guarantee at this stage that the eventual outcome will be favorable.

REFeree REPORTS

Reviewer #1:

The manuscript by Dittmann, Kennedy and colleagues reports that high fat diet (HFD) in mouse models of insulin resistance results in large changes in the phosphotyrosine signaling network. There are a large number of experiments here with almost all based around TMT-based analyses of phosphotyrosine-peptide levels after various treatments. There is also an experiment to look at global protein expression differences in the control vs HFD model. The authors have indeed presented a vastly improved view of diet-induced alterations in tyrosine signaling networks. Finally, they highlight that the majority of the changes are restricted to proteins involved in metabolic processes and pathways.

Overall, this is a meaty and well-written manuscript which presents interesting and complex experimental designs. Often there are replicates and timecourse data in the same experiments. A fraction of the results are also validated via Western blotting. I have only one major comments that would preclude publication.

Major comment:

1) Protein normalization would seem to be an important factor for pY changes measured in this paper. The authors address this in Figure 3. The number of proteins measured here (only 4278) is low. There are at least 10,000 proteins expressed in the liver and so the authors are likely only able to infer what happened in this experiment for half of those. The authors report that "only a small set of proteins (167 of 4278)" were significantly regulated by a HFD. This seems misleading since i) the experiment was powered by just n=2 for control and HFD (4 mice total), and ii) changes that were subthreshold for significance may have still contributed to most of the pY changes seen. It would be more appropriate to normalize by protein and present the comparison of which changes are still significant. There should be new ones as well. This is really just swept under the rug by claiming that most proteins don't change so most pY measurements don't need to be normalized by protein expression. With a paper which relies so heavily on pY measurements, it would really be important to completely wrap up what is happening at the protein level. Note that this is very different from what is commonly done with pY work where an acute treatment leads to changes in just a few hours (or less). In this case the animals have weeks of HFD treatment to produce altered protein levels.

Minor comment:

2) Is figure 4A created as a 9-plex TMT experiment? There are 3 conditions and 3 timepoints. Were there replicates here?

Reviewer #2:

The authors studied H4IIE cells and mice. They treated the H4IIE cells with BSA conjugated either to saturated palmitate or monounsaturated oleate. By immunoblotting they conclude an attenuation by insulin of anti-pAkt-S473 immunoreactivity with cells incubated with palmitate. They looked at ROS with a commercial intracellular ROS kit (MAK145, Sigma-Aldrich) using live cell fluorescence measurements to conclude an effect of palmitate but not oleate to increase ROS. They used quantitative RT-PCR to measure mRNA for G6Pase to show in the absence of insulin an effect of palmitate but not oleate on increased mRNA levels. The authors characterized pY peptides by quantitative LC-MS/MS after labeling with isobaric mass tags. They conclude by hierarchical clustering increases in hundreds of phosphopeptides by palmitate that they conclude correspond to activation of ERK1/2 (ERK2-T183, Y185, ERK1-Y205) and JNK1/2 (JNK1/2-T183, Y185) of MAP kinase cascades as well as of the RTK MET (Y1001), and NCK1-Y105, PI3K28 P85A-Y467 and Y580, PI3K P55G-Y19 as well as SRC family members and SRC family kinase substrates. The authors show immunoblots of increased signals by anti -pT/pY of JNK1/2 and ERK1/2 by palmitate.

Mice livers were studied. The authors fed mice with high fat with or without 2(3)-tert-butyl-4 hydroxyanisole (BHA). The authors assessed glucose tolerance, insulin tolerance and blood glucose levels in the live mice with increases for all with high fat. After liver removal the authors found increased numbers of pY phosphopeptides characterized by LC/MS/MS at 16 weeks of high fat and conclude the increase is in majority in proteins corresponding to enzymes of metabolism. The

authors point out an apparent discrepancy in the reproducibility of results for pY peptides corresponding to the EGF receptor, JAK2, FAK1 and LYN and src family kinases between the 2 experiments summarized in Fig 2D. A marked increase in the actual amount of EGF R is seen by western blot by high fat diet however(Fig3A).

Ion intensities were used to characterize protein abundance from LC/MS/MS. The authors conclude that apolipoproteins, fatty-acid binding proteins, Acetyl-CoA acetyltransferase ACAT1 10 (THIL) and Bile salt activated lipase (CEL) were increased in abundance and 17 members of the cytochrome P450 family were decreased in abundance. They also conclude that metabolic signaling was due to changes in pY levels while RTK signaling of EGFR Jak2 and SFK was due to changes in protein levels.

The authors now administer insulin to mice on the high fat diet. They show decreased insulin stimulated pAKT with high fat diet but regular pY changes in pY levels of the insulin receptor, IRS1 and IRS2 (with some variation) and in Fig4E changes in pY levels of metabolic enzymes.

The authors studied the liver insulin receptor knock out mouse without any insulin injections. They conclude that the KO model showed decreased pY levels of insulin receptor and its downstream targets while EGFR and stat3 pY levels increased in mice with high fat diet.

The authors now return to the H4IIE hepatoma cells in culture now testing the effects of N-acetyl-L-cysteine, butylated hydroxy anisole and rotenone on cells fed with oleate or palmitate. The authors show a principal component analysis of pY levels to conclude that oxidative stress is linked to the altered signaling due to the saturated free fatty acid. They conclude this is due to the inhibitors' effect on src family kinase related signaling pY levels. They also compare the src inhibitor dasatinib. They compare inhibitor effects on insulin signaling and report on G-6-Pase RNA levels to conclude specificity by N-acetyl-L-cysteine on G-6-Pase mRNA as well insulin receptor pY levels.

The authors return to mice.

They add 2(3)-tert-butyl-4 hydroxyanisole(BHA) to the high fat. They show increased pY levels of insulin receptor and decreased pY levels of CEAM1/2 and increased pY levels of glutathione-S transferases. By immunoblot, the authors show lower levels of CEAM1/2 and GSTs. The authors conclude that BHA causes a different signaling network. They also looked at blood glucose, insulin and glucose tolerance to conclude that BHA reverses the negative effects of high fat diet.

The authors' overall conclusion is that high fat leads to rewiring of pY signaling events.

The authors suggest in their Discussion a "diet induced re-wiring of metabolic activities" that SFK inhibition may be a valid strategy to counter diet-induced signaling changes that "decreased ROS burden can globally reverse elevated tyrosine phosphorylation" for insulin resistance. They conclude that BHA "induces effects of unknown consequences".

General remarks:

The observations are of high quality and except for some aspects concerning the EGF receptor and possibly a few other places. The data are consistent over different experimental designs using cultured hepatoma cells and mouse liver. However, if one reads the Discussion of the manuscript, it is difficult to deduce conclusions of a specificity that would enable an increase in understanding of mechanisms that affect liver after high fat diet.

The authors and editors may wish to consider that this is really 3 separate papers with 3 separate sets of advances.

The first paper is the effect of free fatty acids on H4IIE cells.

The second is the effect of high fat on mouse liver.

The third is the effect of oxidative stress inhibitors on H4IIE cells and mouse liver.

This may now enable an audience of cell biologists and endocrinologists to engage with the high-quality data generated by the authors. It may also enable the authors to have more mechanistic conclusions and facilitate a coherent progression in assessment of the data.

All the Western blots are of high quality. At least one of the replicates may be considered for each Figure in the main text and not mainly in the Supplementary Figures. Starting the paper with histograms of ratios of ratios may not be helpful(Fig1A). Starting with the western blot and in the supplementary figures the progression of data for each set of ratios (box and whiskers with data points and not histograms?) may engage the reader to appreciate the quality of the data and rigor of the conclusions.

Again, for the high-quality LC/MS/MS taking the reader through the data coherently may be further engaging and if split among 3 separate papers may be helpful to address mechanistic conclusions. This would mean increasing considerably the data presented as Figures and supplementary Figures in each paper and not just splitting the paper in 3 using only the Figures and supplementary figures already in place.

The discrepancy in the EGF receptor data in mice is puzzling but the Methods indicate that both female and male mice were used. The authors may know that male mouse liver has a much higher abundance of EGF receptor per hepatocyte than female mice. Can the authors indicate in all the data figures with mice which are male, and which are female? The same is true of cytochrome P450s with known female specific ones for example. Were livers from female and males analyzed separately in all cases with data kept separate?

This may be indicated in the supplementary tables. These are of high quality. However, is there a detailed table legend for each of the supplementary tables? This again may help engage and impact the cell biologists and research endocrinologists that such high-quality studies may affect.

Reviewer #3:

Dittman et al. perform global pTyr-proteomic analyses of palmitate vs. oleate-treated HII4E rat hepatocytes and the livers of two cohorts of (differentially) high fat diet (HFD)-fed mice. They identify wide-spread changes in the pTyr-ome, only some of which are due to altered insulin signaling (as revealed by comparisons with the livers from liver insulin receptor knockout (LIRKO) mice. Altered pTyr-proteins include multiple signaling components, particularly RTK and SFK substrates, as well as multiple metabolic enzymes. They find that treatment with the anti-oxidants NAC or BHA or inhibition of SFKs with the ABL/SFK inhibitor dasatanib, have variable effects on these altered pTyr networks in HII4E cells. Similarly, BHA treatment partially reverses the abnormal tyrosine phosphorylation *in vivo*, while normalizing glucose tolerance. The authors present a huge body of work and provide several valuable datasets that will be of benefit to the community. I do, however, have several serious concerns, which mitigate my enthusiasm for publication of the work in its present form, mostly focused on study design. Attention should be given to the following specific points in any revision.

1) The authors carry out two studies of the effects of high fat diet on the liver pTyr-ome. In the first, C57BL6 mice are fed a high fat diet for 6 or 16 weeks or given normal chow NC mice. They then compare the HFD-fed mice to NC-fed mice. They state that they compared each of the HFD-fed mice to "age-matched" normal chow mice. These data can be considered a "test set." They then carry out another study using C57BL6 mice bearing *Lgr5-ERTamCre* and *Rosa26lacZ* transgenes, fed a high fat diet for 12 or 30 weeks, again compared with NC-fed mice. These could be viewed as an attempt at a "validation set," yet there are serious problems here:

a) Although the text states that comparisons were made to age-matched NC controls, in the heat map in Figure 2, NC-30 data are shown. Why are mice from the test set compared with NC from the validation set? What is the variation between NC samples at 6 weeks, 12 weeks, 16 weeks and 30 weeks? Why aren't comparisons made to the matched samples?

b) Even more problematic, of course, is that the "validation set" is performed on different mice, housed at different centers, and fed HFD for different times! Although the authors might think that the *Lgr5* and *Rosa* transgenes do not affect outcomes, they don't know this. In any event, how is one to interpret the differences in data obtained? Are these time of HFD-dependent? UMass Worcester vs MIT housing dependent? Differences in strain-dependent? (I think it is highly likely, given current practices in most laboratories, that the *Lgr5/Rosa* mice were intercrossed extensively before use-and then assumed to be "C67BL/6," when they are probably edging towards a new strain).

2) A similar problem attends the comparisons with the LIRKO mice, with the additional concern that instead of using *fl/fl* *Insr* mice as the control, *Alb-Cre* mice should have been used (*Cre* is well-documented to cause DNA damage, for example).

3) Given these concerns, the authors should at the very least perform a real validation set or remove the "Study 2" data entirely.

4) Similarly, where were the 16-week HFD-fed Alb-Cre samples compared with NC 30 samples (Figure 5, legend)? Again, it would seem that comparisons are made to "unmatched" samples.

5) The authors note, referring to Supp Fig 5A that "NAC was similarly effective in reducing PA-induced tyrosine phosphorylation as BHA. Although the data support the conclusion that both anti-oxidants decrease overall pTyr, it is equally notable that they also have distinct effects on different groups of pTyr proteins. The authors should discuss how the two anti-oxidants act and why they might affect different pTyr proteins.

6) Left unclear is any sense of which events are upstream and which are consequences. For example, what does Dasatanib treatment do to ROS? What does JNK activation do to ROS? What do each of the anti-oxidants do to SFKs/JNK? Obviously, the authors do not have to resolve the entire workings of the rewired signaling network but a few general principles would be helpful and would increase the impact of their work.

7) The authors mention a recent proteomic study by Farese (Li et al., MCP 2018). Comparison of their data to the previous study seems warranted.

8) Minor points:

a) The authors should do a better job of punctuating their manuscript. For example, there seems to be a general aversion to commas, which makes parsing some sentences quite difficult.

b) Sentences should not begin with numbers (i.e., "8-week old" should be "Eight-week old").

c) GTT means "glucose tolerance test" not "glucose tolerant test."

1st Revision - authors' response

17 May 2019

Reviewer 1:

1. Protein normalization would seem to be an important factor for pY changes measured in this paper. The authors address this in Figure 3. The number of proteins measured here (only 4278) is low. There are at least 10,000 proteins expressed in the liver and so the authors are likely only able to infer what happened in this experiment for half of those.

Response: While we agree that the number of detectable proteins in the liver may be larger, the number of proteins detected and quantified in our study compares favorably with other studies of the liver proteome, which report between 965-6000 proteins identified. The discrepancy in these numbers can be explained by the use of different mass spectrometry platforms, instrument performance, differences in sample preparation and differences in data processing and peptide filtering. Here we have applied a relatively stringent filter for peptide identification, including a high cut-off based on search engine score (≥ 25). With these settings, we obtain ~5200 proteins with a target decoy false discovery rate (FDR) < 0.003 , lower than the 0.01 (1%) typically used as a filter by many other studies. In addition, we require at least two unique peptides or three peptide-to-sequence matches (psms) per protein to minimize protein FDR. Applying these filters decreases the number of proteins that can be reliably quantified to 4459, but we feel that these filters make identification and quantification as robust as possible, with the goal of providing a set of high-confidence IDs as potential follow up candidates for other scientists.

The authors report that "only a small set of proteins (167 of 4278)" were significantly regulated by a HFD. This seems misleading since i) the experiment was powered by just $n=2$ for control and HFD (4 mice total), and ii) changes that were subthreshold for significance may have still contributed to most of the pY changes seen. It would be more appropriate to normalize by protein and present the comparison of which changes are still significant. There should be new ones as well. This is really just swept under the rug by claiming that most proteins don't change so most pY measurements don't need to be normalized by protein expression. With a paper which relies so heavily on pY measurements, it would really be important to completely wrap up what is happening at the protein level. Note that this is very different from what is commonly done with pY work where an acute treatment leads to changes in just a few hours (or less). In this case the animals have weeks of HFD treatment to produce altered protein levels.

Response: We agree that the duration of the treatment can lead to significant changes in protein levels and that it is important to dissect whether these alterations are underlying the phosphorylation response as opposed to changes in signaling. **We have tried to address this issue as thoroughly as possible by including an additional replicate for both conditions and added significance levels to Figure EV 3A.** Incorporating the third replicate now provides additional statistical power, and **thus changes the number of significantly changing proteins to 642 out of 4459 reliably quantifiable proteins.** The changes are reflected in the figure legends, methods section, and the statistical analysis. Please note that we have changed our determination of significance and added more information in Figure 3B, the methods section and figure legends. We have amended the results and discussion sections accordingly. In addition, we included a second figure display (Figure EV3B) that captures the direction of expression and phosphorylation of proteins that share common phosphopeptides and therefore could not be displayed in Figure EV3A. We would also like to highlight that for 99 out of the 111 phosphotyrosine sites that were found to be significantly changed by diet, we were able to generate protein expression profiles across these two conditions. We believe, that by limiting our conclusions about the effect of protein expression on phosphorylation to those proteins, we have captured the most important aspects. We feel that these changes have strengthened the conclusions in the manuscript, as we can more accurately state that the vast majority of the signaling network changes are not due to changes in protein expression, but rather due to changes in protein post-translational modifications.

2. Minor comment:

Is figure 4A created as a 9-plex TMT experiment? There are 3 conditions and 3 timepoints. Were there replicates here?

Response: Yes, there were replicates present in this analysis. We clarified the number of replicates per condition in the figure legend. The significance test in Figure 4B-C takes the number of replicates into account.

Reviewer 2:

1. The observations are of high quality and except for some aspects concerning the EGF receptor and possibly a few other places. The data are consistent over different experimental designs using cultured hepatoma cells and mouse liver. However, if one reads the Discussion of the manuscript, it is difficult to deduce conclusions of a specificity that would enable an increase in understanding of mechanisms that affect liver after high fat diet.

The authors and editors may wish to consider that this is really 3 separate papers with 3 separate sets of advances.

The first paper is the effect of free fatty acids on H1IE cells.

The second is the effect of high fat on mouse liver.

The third is the effect of oxidative stress inhibitors on H1IE cells and mouse liver.

This may now enable an audience of cell biologists and endocrinologists to engage with the high- quality data generated by the authors. It may also enable the authors to have more mechanistic conclusions and facilitate a coherent progression in assessment of the data.

Response: Thank you for this suggestion. We discussed splitting the manuscript and decided against it. We have made changes to the discussion to more effectively communicate new mechanistic insights that can be deduced from the data and added more information to figure and table legends to facilitate assessment. We believe that the strength of this work lies in the generalizability of the effects of lipid overload on the pTyr proteome across different systems. We think that separating the manuscript would create too many gaps in the separate papers that other reviewers might point out as weaknesses.

2. All the Western blots of are high quality. At least one of the replicates may be considered for each Figure in the main text and not mainly in the Supplementary Figures. Starting the paper with histograms of ratios of ratios may not be helpful (Fig1A). Starting with the western blot and in the supplementary figures the progression of data for each set of ratios (box and whiskers with data points and not histograms?) may engage the reader to appreciate the quality of the data and rigor of the conclusions.

Response: We agree with this suggestion and have changed Figure 1A accordingly. The manuscript now contains Western blots where appropriate in the main figures and we included individual data points rather than averages in the summary figures.

- Again, for the high-quality LC/MS/MS taking the reader through the data coherently may be further engaging and if split among 3 separate papers may be helpful to address mechanistic conclusions.

This would mean increasing considerably the data presented as Figures and supplementary Figures in each paper and not just splitting the paper in 3 using only the Figures and supplementary figures already in place.

Response: While we appreciate the added space that three manuscripts would offer, we feel that dividing the story into three separate manuscripts would cause a loss of the systems-level information that is gained by quantifying changes in vitro and in vivo within a single manuscript.

- The discrepancy in the EGF receptor data in mice is puzzling but the Methods indicate that both female and male mice were used. The authors may know that male mouse liver has a much higher abundance of EGF receptor per hepatocyte than female mice. Can the authors indicate in all the data figures with mice which are male, and which are female? The same is true of cytochrome P450s with known female specific ones for example.

Response: Thank you for pointing out these differences, we have now included it in the manuscript as a possible explanation for the discrepancies in EGFR between study 1 and 2. All figure legends and tables now include more detailed study information.

- Were livers from female and males analyzed separately in all cases with data kept separate?

Response: Yes, they were kept separate. We have now included that information in figure and table legends more explicitly.

- This may be indicated in the supplementary tables. These are of high quality. However, is there a detailed table legend for each of the supplementary tables? This again may help engage and impact the cell biologists and research endocrinologists that such high-quality studies may affect.

Response: We have added a detailed legend to the top of each table to better connect each table to each study that was performed. We hope this additional information will facilitate the accessibility of these tables to the broader community.

Reviewer 3:

The authors carry out two studies of the effects of high fat diet on the liver pTyr-ome. In the first, C57BL6 mice are fed a high fat diet for 6 or 16 weeks or given normal chow NC mice. They then compare the HFD-fed mice to NC-fed mice. They state that they compared each of the HFD-fed mice to "age-matched" normal chow mice. These data can be considered a "test set." They then carry out another study using C57BL6 mice bearing Lgr5-ERTamCre and Rosa26lacZ transgenes, fed a high fat diet for 12 or 30 weeks, again compared with NC-fed mice. These could be viewed as an attempt at a "validation set," yet there are serious problems here:

- Although the text states that comparisons were made to age-matched NC controls, in the heat map in Figure 2, NC-30 data are shown. Why are mice from the test set compared with NC from the validation set? What is the variation between NC samples at 6 weeks, 12 weeks, 16 weeks and 30 weeks? Why aren't comparisons made to the matched samples?

Response: We apologize for the confusion caused by our prior naming scheme. In fact, NC30 in Figure 2A refers to the normalization sample from study 1 (24 weeks NC with 30 minutes of insulin stimulation) that was included in each MS analysis of study 1 in order to compare relative fold-changes from different runs with each other. It is not the 30-week age-matched NC from study 2. **We have now renamed the normalization samples for study 1 and 2 in the manuscript and all figure legends.** For study 1 there is one NC that is age-matched with HFD6 and HFD16 (all 24 weeks when sacrificed) by starting the high-fat diets in a staggered fashion, i.e. after 8 and 18 weeks for 16 and 6 weeks HFD respectively. We have reworded this part in the methods section and figure legends. Please refer to Figure EV2D and Dataset EV3 for a comparison of NC12 and NC30. Because of

the way the analyses were set up we cannot make comparisons between NC from study 1 and NC12 and NC30 from study 2.

- Even more problematic, of course, is that the "validation set" is performed on different mice, housed at different centers, and fed HFD for different times! Although the authors might think that the Lgr5 and Rosa transgenes do not affect outcomes, they don't know this. In any event, how is one to interpret the differences in data obtained? Are these time of HFD-dependent? UMass Worcester vs MIT housing dependent? Differences in strain-dependent? (I think it is highly likely, given current practices in most laboratories, that the Lgr5/Rosa mice were intercrossed extensively before use-and then assumed to be "C67BL/6," when they are probably edging towards a new strain).

Response: We agree that the 'test' and 'validation' sets are not perfectly matched, and yet this is what makes the conclusions most robust – despite the differences in housing, background, sex, and diet, placing mice on extended HFD leads to dysregulated tyrosine phosphorylation signaling networks in their respective mouse livers, including increased phosphorylation of metabolic signaling network proteins and RTK signaling. We now include more details in the discussion on explaining the rationale for choosing different experimental designs.

- A similar problem attends the comparisons with the LIRKO mice, with the additional concern that instead of using fl/fl Insr mice as the control, Alb-Cre mice should have been used (Cre is well-documented to cause DNA damage, for example).

Response: Conditional mouse knockout studies can employ two controls: the Flox control (because of possible hypomorphic effects of the Floxed allele) and the Cre control (because of possible Cre-mediated responses, including DNA damage). The multiplexing strategy using biological replicates for mass spectroscopy only allows pair-wise comparisons because of the limitations of the total number of samples that can be multiplexed. We therefore needed to choose one control - either Flox or Cre. We chose to employ the Flox control because these mice are littermates of the liver-specific KO mice. The Cre control would require the use of non-littermate mice, an experimental design that is sub-optimal.

- Given these concerns, the authors should at the very least perform a real validation set or remove the "Study 2" data entirely.

Response: Please refer to our response to point 2. In addition, we now include more details on study design and sex in the figure and table legends to make this information more readily available to the reader.

- Similarly, where were the 16-week HFD-fed Alb-Cre samples compared with NC 30 samples (Figure 5, legend)? Again, it would seem that comparisons are made to "unmatched" samples.

Response: Please refer to our response to point 1 explaining the inadvertently confusing naming. We have now changed the display to showing levels in HFD16 relative to NC.

- The authors note, referring to Supp Fig 5A that "NAC was similarly effective in reducing PA-induced tyrosine phosphorylation as BHA. Although the data support the conclusion that both anti-oxidants decrease overall pTyr, it is equally notable that they also have distinct effects on different groups of pTyr proteins. The authors should discuss how the two anti-oxidants act and why they might affect different pTyr proteins.

Response: Thank you for this suggestion. We have now elaborated on possible explanations for the differences and added it to the discussion.

- Left unclear is any sense of which events are upstream and which are consequences. For example, what does Dasatinib treatment do to ROS? What does JNK activation do to ROS? What do each of the anti-oxidants do to SFKs/JNK? Obviously, the authors do not have to resolve the entire workings of the rewired signaling network but a few general principles would be helpful and would increase the impact of their work.

Response: We have added experimental data examining the effects of Dasatinib on ROS as Figure EV5G. Based on these data we conclude that inhibiting SFK activity has no effect on ROS levels, suggesting that ROS are upstream of changes in SFK substrate phosphorylation sites. The effects of NAC and BHA on SFKs and JNK can be extracted

from Dataset EV6 showing that NAC and BHA reduce phosphorylation on JNK2 and selected SFK substrates (mentioned in the results section), but not the SFK autophosphorylation site. We conclude that selected sites in the SFK signaling network are affected by ROS levels but not necessarily through changes in SFK activity, potentially indicating a role for ROS-mediated changes in protein tyrosine phosphatase activity in regulating these phosphorylation sites. We have included this aspect of the results in the discussion.

8. The authors mention a recent proteomic study by Farese (Li et al., MCP 2018). Comparison of their data to the previous study seems warranted.

Response: Thank you for this suggestion. Unfortunately, the work by Li et al (MCP 2018) focused on a thorough characterization of S/T phosphorylation in the context of insulin stimulation with and without palmitate. Given the differences in experimental design and phosphopeptide enrichment strategies, i.e. global phosphorylation vs pTyr enrichment, the number of pTyr peptides in the Li et al study was significantly smaller than in our study and does not allow for a meaningful comparison of the two studies.

Minor points:

9. The authors should do a better job of punctuating their manuscript. For example, there seems to be a general aversion to commas, which makes parsing some sentences quite difficult.

Response: We have made changes to the manuscript where appropriate

10. Sentences should not begin with numbers (i.e., "8-week old" should be "Eight-week old").

Response: We altered this sentence and checked the manuscript to avoid any other occurrences.

11. GTT means "glucose tolerance test" not "glucose tolerant test."

Response: We fixed this error.

2nd Editorial Decision

27 June 2019

Thank you for sending us your revised manuscript. We have now heard back from the two referees who were asked to evaluate your study. As you will see below, the reviewers are satisfied with the modifications made and think that the study is now suitable for publication.

Before we formally accept the study for publication we would ask you to address minor editorial issues.

REFeree REPORTS

Reviewer #1:

The authors have adequately addressed my previous concerns.

Reviewer #3:

The authors have addressed my major concerns, and the revised paper is suitable for publication

Corresponding Author Name: Forest White

Manuscript Number: